# Breaking the capacity bottleneck of lithium-oxygen batteries through reconceptualizing transport and nucleation kinetics

Zhuojun Zhang [1], Xu Xiao [1] ✉, Aijing Yan[1], Kai Sun[1], Jianwen Yu[1] & Peng Tan [1,2] ✉

The practical capacity of lithium-oxygen batteries falls short of their ultra-high theoretical value. Unfortunately, the fundamental understanding and enhanced design remain lacking, as the issue is complicated by the coupling processes between $Li_2O_2$ nucleation, growth, and multi-species transport. Herein, we redefine the relationship between the microscale $Li_2O_2$ behaviors and the macroscopic electrochemical performance, emphasizing the importance of the inherent modulating ability of $Li^+$ ions through a synergy of visualization techniques and cross-scale quantification. We find that $Li_2O_2$ particle distributed against the oxygen gradient signifies a compatibility match for the nucleation and transport kinetics, thus enabling the output of the electrode's maximum capacity and providing a basis for evaluating operating protocols for future applications. In this case, a 150% capacity enhancement is further achieved through the development of a universalizing methodology. This work opens the door for the rules and control of energy conversion in metal-air batteries, greatly accelerating their path to commercialization.

Lithium-oxygen batteries (LOBs), with significantly higher energy density than lithium-ion batteries, have emerged as a promising technology for energy storage and power[1-4]. Research on LOBs has been a focal point, showing great potential for high-rate performance and stability[1,5-7]. Despite significant advancements in various aspects, practical LOBs have yet to fully realize their innate potential for ultra-high theoretical energy[8,9].

Indeed, when all pores of the air electrode are filled with metal oxides, the electrode reaches its theoretical capacity. Unfortunately, the spatial utilization of the air electrode is often insufficient, resulting in an actual output capacity that falls far short of the theoretical value. It is challenging to assess whether an output capacity has reached the upper limit under the current operating protocol, making targeted regulation confusing. This dilemma raises questions and concerns, what are the indicators that an electrode has reached its actual maximum capacity? How can the capacity limit be further enhanced? These are crucial for breaking the capacity bottleneck of LOBs.

The key lies in establishing the relationship between the macroscopic electrochemical performance and the $Li_2O_2$ microscopic behaviors, such as nucleation, morphology, and distribution. Due to the complexity of physicochemical processes involving the coupling of phase transitions, mass transfer, and faradaic processes, a unified and clear understanding of nanoscale behavior is still lacking. The $Li_2O_2$ morphology plays a fundamental role in determining the actual capacity[10]. Several groups stated that a thin $Li_2O_2$ film can hinder electron transfer and prematurely terminate discharge[11-13]. Although the $Li_2O_2$ properties can be significantly changed by applying solid or liquid-phase catalysts[14,15], the underlying nucleation-growth theory remains controversial. For the transport kinetics, disruption of transport in gas or ion will also lead to the reaction cessation. In traditional perception, solid products predominantly accumulate on the oxygen side, aligning with the gas gradient at low saturation levels[16-18]. Due to all processes occurring in 3D space, it is essential to obtain more detailed data for the interactions of multiple species in the direction of

[1]Department of Thermal Science and Energy Engineering, University of Science and Technology of China (USTC), Hefei, Anhui, China. [2]State Key Laboratory of Fire Science, University of Science and Technology of China (USTC), Hefei, Anhui, China. ✉e-mail: xiaoxu@ustc.edu.cn; pengtan@ustc.edu.cn

electrode thickness, rather than focusing solely on individual sites or faces.

Another factor contributing to the limited understanding of microscopic behaviors is the constraints in research methods. Advanced observation techniques, such as in-situ scanning electron microscopy (SEM)[19,20], transmission electron microscopy (TEM)[21–24], and atomic force microscopy (AFM)[25–27], have been widely used to provide timely images. However, they can only characterize the $Li_2O_2$ on the surface level (cathode top or bottom) and fail to reveal the distribution and morphology of $Li_2O_2$ inside the porous electrode. The lack of experimental data inside the real air electrode makes it challenging to infer the species distribution. A usual method to quantify the mass transport within the porous media is to develop multi-field models[28–30]. Although the multidimensional macroscopic continuum models have been employed to calculate the active species concentration and $Li_2O_2$ thickness, they are insufficient in portraying $Li_2O_2$ nucleation, morphology, and interface evolution[31–33]. Regrettably, few models are yet available to provide a good quantitative understanding of the mesoscale processes[34,35]. Therefore, it is urgent to utilize the cross-scale simulation and multi-angle observations to reveal the phase transition process in porous electrodes.

In this work, to exclude the possible influence of sensitive terms (e.g., donor number, catalyst activity) on the $Li_2O_2$ behaviors, all mechanistic investigations are conducted with fixed components. The inherent modulating ability of $Li^+$ ions is utilized to alter the initial kinetic characteristics. A visualized air electrode is constructed, and multi-filed cross-scale modeling combining mesoscopic phase field and macroscopic continuum medium methods is established, based on which a mechanistic understanding of $Li_2O_2$ nucleation and distribution is presented. We directly show and confirm that the $Li_2O_2$ particle distributed against the oxygen gradient is the result of a trade-off between nucleation and transport kinetics. Further, the revealed mechanism achieves a significant 150% increase in maximum capacity by adopting a universalizing methodology. This work provides a valuable advance in the knowledge of laws and control of capacity in metal-air battery systems, thereby greatly promoting their practical process.

## Results and discussion
### Hidden electrochemical behaviors
To facilitate the comprehensive analysis of the solid-liquid interface, $Li_2O_2$ distribution, and mass transport inside the electrode, an integrated carbon-coated anodic aluminum oxide (C-AAO) air electrode as the visualized electrode is constructed in this work. This electrode is porous, consisting of vertically penetrating channels with diameters of 390 nm (Supplementary Fig. 1). The electrolyte is configured with lithium bis (tri-fluoromethane sulfonyl) imide (LiTFSI) and tetra-ethylene glycol dimethyl ether (TEGDME) in proportion. By employing electrolytes of varying concentrations, the balance between $Li^+$ ions and oxygen, and the transport ability of electrolytes are regulated effectively.

To ensure the universality of the conclusion, the electrochemical behaviors of disordered (a general carbon nanotube electrode) and visualized electrodes are compared. The selection of applied current is based on the actual electrochemical active area of the electrodes to address the effects of structural differences. $0.1\,mA\,cm^{-2}$ applied in disordered electrodes is equal to $300\,mA\,g^{-1}$ of C-AAO electrodes according to the double-layer capacitance ($C_{dl}$) method (Supplementary Fig. 2 and Supplementary Table 1), thereby resulting in similar absolute capacities at the same $Li^+$ ion concentration (Supplementary Table 2). Figure 1a–c shows the voltage-capacity curves with $0.05–2\,M$ electrolytes. The capacity exhibits an increase and subsequent decrease with an increase of the $Li^+$ ion concentration (Fig. 1c). The maximum capacity occurs at $0.5\,M$, and the capacity of $1\,M$ electrolyte is higher than that of $0.1\,M$ electrolyte. Ionic conductivity is commonly

considered a general explanation for capacity variations, with the reported peak conductivity of LiTFSI/TEGDME occurring between $1\,M$ and $2\,M$[36,37]. However, the $Li^+$ ion concentration for maximum capacity does not coincide with that for peak ionic conductivity. Thus, ionic conductivity is not the sole determinant of electrochemical performance, and what else happens accompanied?

Interestingly, the initial voltage plateau (the starting point of the voltage plateau, which is quantitatively determined by the peak of dV/dQ plots in Supplementary Figs. 3, 4) does not conform to the trend of discharge capacity, which confirms the non-uniqueness of influencing factors. The initial voltage plateaus are shown in Fig. 1d, e, and the trend is summarized in Fig. 1f. Specifically, the initial voltage plateau decreases as the $Li^+$ ion concentration increases for both disordered and visualized electrodes. The $Li^+$ ion concentration of $0.05\,M$ exhibits the highest initial voltage plateau and, paradoxically, demonstrates the lowest capacity. Therefore, the initial voltage plateau appears to be more closely related to the viscosity or oxygen solubility of the electrolyte rather than the ionic conductivity.

The behaviors of ohmic impedance ($R_s$) and charge transfer impedance ($R_{ct}$) exhibit significant dependence on the $Li^+$ ion concentration. The results of electrochemical impedance spectroscopy (EIS) of disordered and visualized electrodes at the fixed capacities are shown in Fig. 1g, h, and the increase in impedance is shown in Fig. 1i and Supplementary Table 3. The criteria for selecting the fixed capacity is to do so before the rapid voltage drop. The net $R_{ct}$ reaches a maximum in the $0.1\,M$ electrolyte (2-4-fold) while remaining at lower and similar levels in the $0.5–2\,M$ electrolytes. Notably, $R_s$ exhibits negligible growth in the $0.5–2\,M$ electrolytes but increases substantially in the $0.1\,M$ electrolyte (~ 20-fold). Of course, it is undeniable that a low $Li^+$ ion concentration does lead to higher values of $R_s$ and $R_{ct}$ before discharge (Fig. 1g, h), which may affect discharge to some extent. Since the above electrochemical behaviors cannot be explained completely by the physical properties of electrolytes and are inconsistent with each other, we thus turn to nucleation and transport kinetics.

### $Li_2O_2$ nucleation-growth theory
The SEM images of $Li_2O_2$ morphologies on the disordered electrode, both at full discharge and at a fixed capacity, are shown in Supplementary Figs. 5 and 6. In the $0.1\,M$ electrolyte, the carbon nanotubes (CNTs) are enveloped by a film-like layer (Supplementary Figs. 5a, 6a). In the $0.5–2\,M$ electrolytes, despite the generation of numerous $Li_2O_2$ particles on the electrode, some CNTs can still be exposed in the electrolyte (Supplementary Fig. 5b and 6b, d). The peaks of X-ray diffraction (XRD) patterns at $32.9°$, $35.0°$, and $58.7°$ indicate the formed $Li_2O_2$ is crystalline (Supplementary Fig. 7). Similarly, on the top of the visualized electrode, the $Li_2O_2$ morphology undergoes a transition from a 3D film-like structure at $0.05\,M$ to a mixed state of films and particles at $0.1\,M$ and eventually becomes particle-like when the $Li^+$ ion concentration exceeds $0.5\,M$, as shown in Fig. 2a–d and Supplementary Fig. 8. The above findings demonstrate consistency in both electrochemical and $Li_2O_2$ behaviors.

The impedance behaviors can be attributed to the evolution of $Li_2O_2$ morphologies. In the $0.1\,M$ electrolytes, the electron transport and the reactivity of reaction sites are significantly inhibited by the electrode passivation caused by the crystalline $Li_2O_2$ film. The former typically corresponds to the relationship $\sigma = 10^7 \sum \delta^\tau$ based on the electron tunneling effect[11,35]. $\sigma$ is the conductivity of the $Li_2O_2$ film, $\delta$ is the film thickness, and $\tau$ is the coefficient ($-17.18$)[11,35]. In the $0.5–2\,M$ electrolytes, the contribution of $Li_2O_2$ particles to $R_s$ may not adhere to the aforementioned equation. The findings provide a guide for understanding the role of $Li_2O_2$ morphology in the modeling, which has been subsequently incorporated into the model assumptions in this work.

To better elucidate the formation of $Li_2O_2$ film in the electrolyte with low $Li^+$ ion concentration, a nucleation-growth mechanism of

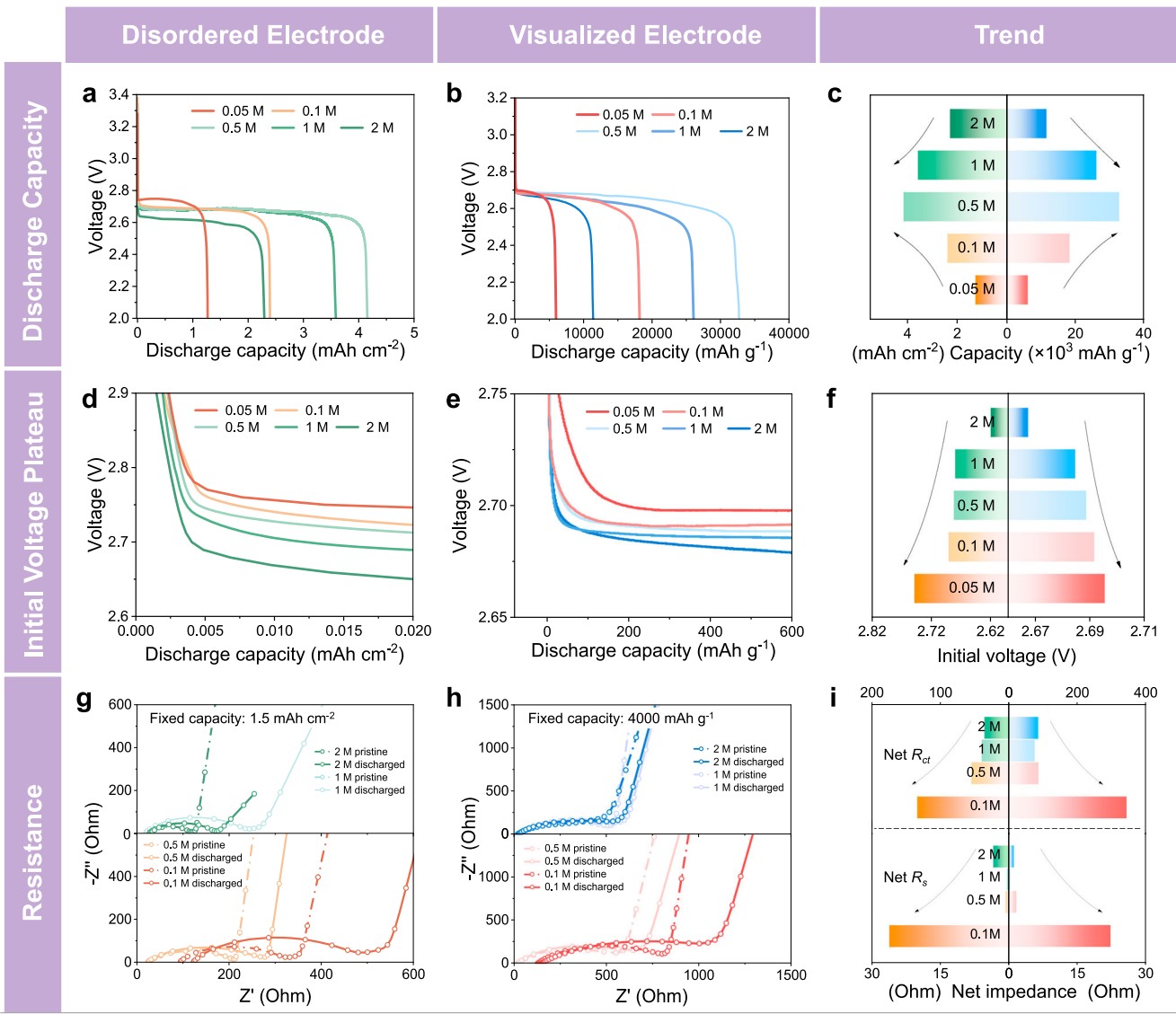

**Fig. 1 | Electrochemical behaviors under different Li+ ion concentrations.** Galvanostatic discharge curves with 0.05–2 M electrolytes using (**a**) disordered electrodes at 0.1 mA cm$^{-2}$ and (**b**) visualized electrodes at 300 mA g$^{-1}$. **c** The trend of discharge capacity with Li+ ion concentration. Initial voltage plateau using (**d**) disordered electrodes and (**e**) visualized electrodes. **f** Trend of initial voltage plateau

with Li+ ion concentration. EIS results using (**g**) disordered electrodes at a fixed capacity of 1.5 mAh cm$^{-2}$ and (**h**) visualized electrodes at a fixed capacity of 4000 mAh g$^{-1}$. **i** Comparison of the net $R_s$ and net $R_{ct}$, where net $R$ is defined as $R_{discharged}$ - $R_{pristine}$.

Li$_2$O$_2$ is proposed. The nucleation of Li$_2$O$_2$ is distinct from the deposition of metallic Li or Zn in metal-based batteries. Specifically, Li deposition is a single-step Faraday reaction (Li+ + e$^-$ → Li), while Li$_2$O$_2$ formation involves multiple steps of electrochemical and chemical reactions. When discussing nucleation, overpotential is a critical factor. It is widely acknowledged that a sudden nucleation overpotential drives Li to nucleate instantaneously, followed by a decrease in overpotential and stabilization during the Li growth stage[38–40]. In the LOBs, however, the overpotential increases monotonically since it arises from the kinetics from oxygen to superoxide (O$_2$ + e$^-$ → O$_2^-$), rather than a direct phase transition. The decrease in salt concentration enhances the oxygen solubility[41,42], while simultaneously freeing up sites on the electrode surface for oxygen molecules to absorb before discharge (Fig. 2e). Therefore, a negative correlation between the initial voltage plateau and Li+ ion concentration in Fig. 1d–f is observed (Fig. 2f) and the early voltage is controlled by the adsorbed oxygen (Fig. 2g). Additionally, more nuclei are generated at the beginning of discharge in the presence of higher oxygen concentration and faster

electrode kinetics in the electrolytes with low Li+ ion concentration (Fig. 2f). The initial nuclei density will further determine the product morphology, thereby influencing the voltage characteristics in the later stage of discharge (Fig. 2g).

The effect of the nuclei density on the Li$_2$O$_2$ morphology and overpotential is further revealed via quantitative calculation using a phase field model. At 0.1 M, more adsorbed LiO$_2$ is formed initially, as illustrated in Fig. 2e, which results in more Li$_2$O$_2$ nuclei. With Li$_2$O$_2$ growth, adjacent nuclei are prone to interconnect during the early discharge stages, eventually forming a film-like structure (Fig. 2h). In contrast, the nuclei density is decreased in electrolytes with higher Li+ ion concentrations, resulting in particle-like Li$_2$O$_2$ formations (Fig. 2i). The overpotential can be expressed by the modified Tafel equation considering the coverage ($\theta$) of Li$_2$O$_2$ on the electrode surface.

$$\eta = \frac{2.3RT}{\alpha nF} \lg\left(\frac{i}{i_0} \cdot \frac{1}{1-\theta}\right) \quad (1)$$

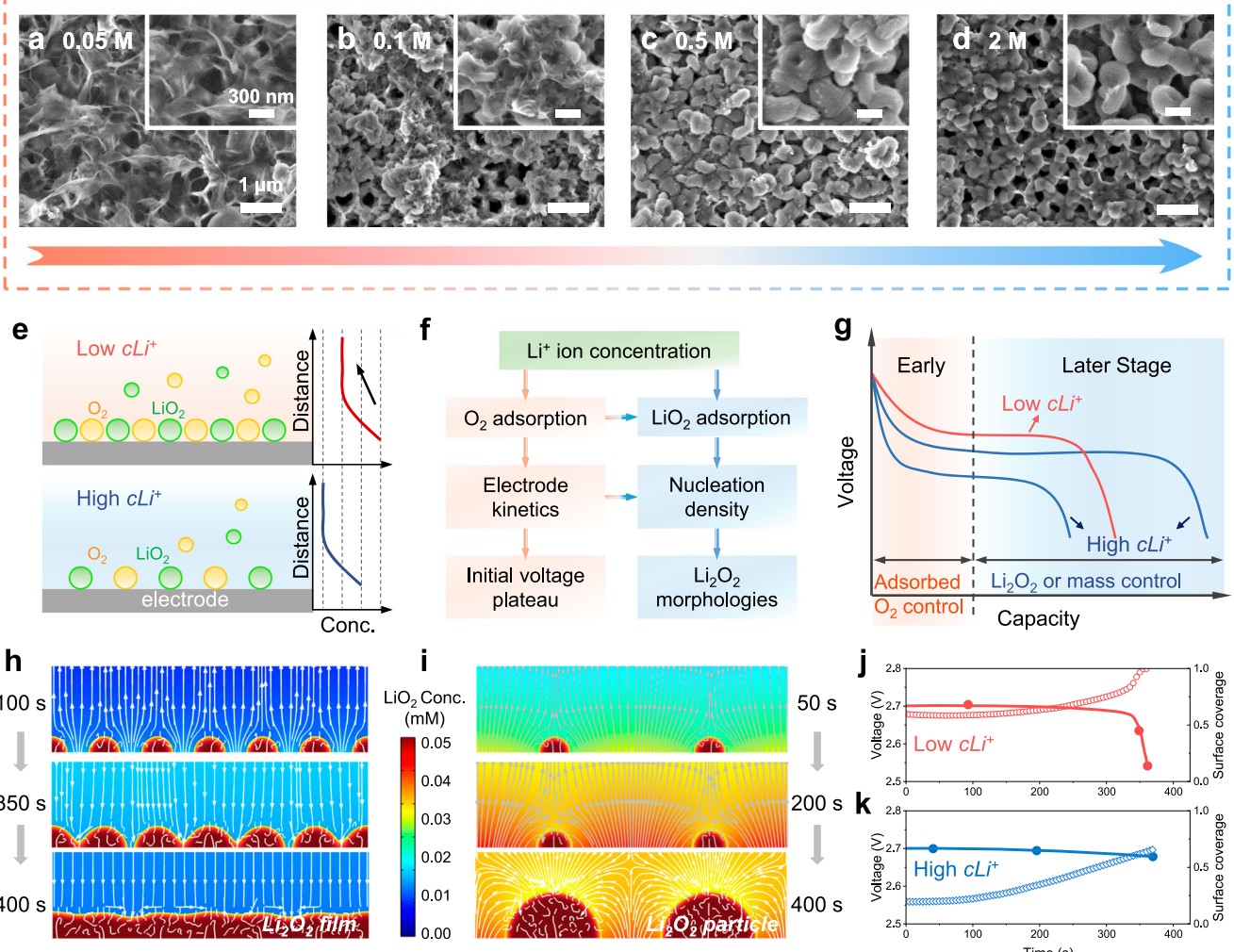

**Fig. 2 | Li$_2$O$_2$ nucleation-growth theory and the relationship between product morphology and overpotential. a–d** The evolution of Li$_2$O$_2$ morphologies at full discharge with different Li$^+$ ion concentrations. The scale bars of the large image are 300 nm, while that of the small image are 1 μm. **e** Scheme of adsorption/distribution of O$_2$ and LiO$_2$ at the solid-liquid interface in electrolytes with different concentrations. **f** The mechanism of Li$^+$ ion concentration on initial voltage plateau and nucleation. **g** Scheme of voltage characteristics and control factors at early and later stages. The growth evolution of (**h**) Li$_2$O$_2$ film with low cLi$^+$ and (**i**) particle with high cLi$^+$ simulated by the field phase method. The relationship between coverage of Li$_2$O$_2$ on the electrode surface and discharge voltage with (**j**) low cLi$^+$ and (**k**) high cLi$^+$.

where $i$ is the applied current and $i_O$ is the exchange current density. In the low Li$^+$ ion concentration electrolytes, when the electrode is covered by a Li$_2$O$_2$ film, the voltage drops rapidly (Fig. 2j). Due to the electrode passivation, the LiO$_2$ decreases in the later discharge stage (Fig. 2h). Conversely, in the electrolyte with high Li$^+$ ion concentration, since the electrode surface remains exposed, the voltage plateau remains essentially stable with increasing LiO$_2$ (Fig. 2k). Therefore, the electrode passivation caused by the film-like structure is an important factor contributing to battery failure with low Li$^+$ ion concentrations (Supplementary Fig. 9). It has been preliminarily verified that the nucleation-growth theory and the failure mechanism in electrolytes with low Li$^+$ ion concentration can be extended to high donor-number electrolyte systems (Supplementary Fig. 10).

## Quantitative analysis of transport kinetics

Although the nucleation-growth theory provides a failure mechanism at low Li$^+$ ion concentrations, the factors limiting capacity with high Li$^+$ ion concentrations remain unclear, particularly why the maximum occurs at 0.5 M. Due to the brittleness of the C-AAO electrode, the morphology and distribution of Li$_2$O$_2$ can be well retained when it is snapped, which further enables visualization inside the gas electrode.

This highly consistent array structure provides clear and accurate transport pathways and species flux, allowing for channel unit separation to enable multi-physics field modeling. Therefore, the synergy of visualization for the porous electrode interior and 3D heterogeneous modeling is used to quantitatively understand the electrochemical behaviors and transport kinetics. The selection and rationalization analysis of transport kinetics parameters in electrolytes with different Li$^+$ ion concentrations is detailed in Supplementary Fig. 11 and Supplementary Table 4.

Figure 3a shows the cross-section of the visualized electrode, where the positions from the oxygen side to the middle part and then to the separator side are observed sequentially. Supplementary Figs. 12 and 13 suggest that Li$_2$O$_2$ film is dominant in the 0.05 M and 0.1 M electrolytes. The film terminates discharge before channel space is fully utilized. While in the 0.5 M electrolyte, the situation is significantly different. The visible Li$_2$O$_2$ particles aggregate and fill the channels, as shown in Fig. 3b. It can be inferred that the lack of storage space for solid products or slowing transport caused by the increasing tortuosity becomes limiting factors. Interestingly, the channels are almost empty again when Li$^+$ ion concentration further increases to 1 M and 2 M.

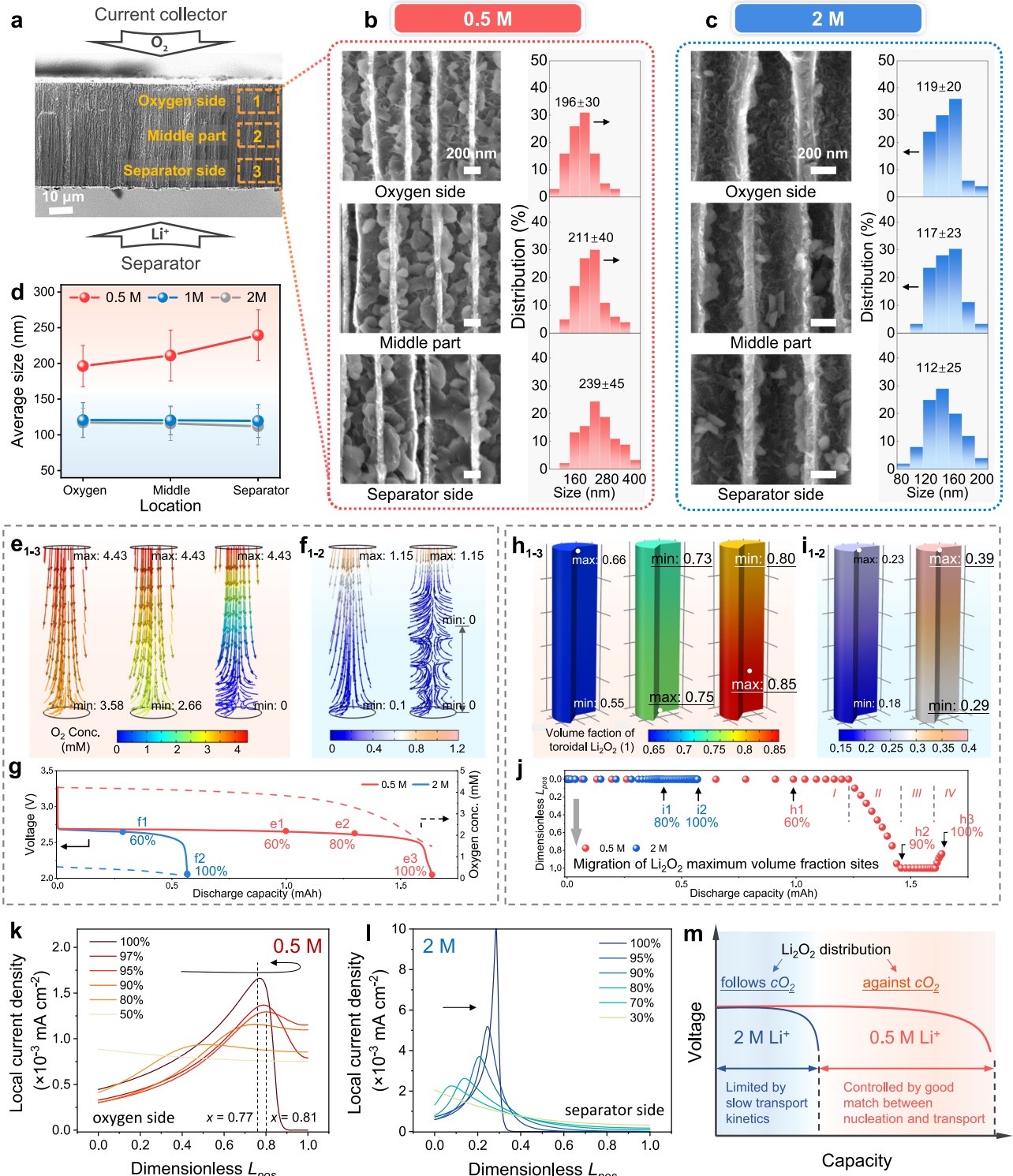

**Fig. 3 | Quantitative analysis of species transport inside visualized electrodes under different Li⁺ ion concentrations. a** The cross-section of the C-AAO electrode with a channel diameter of 390 nm before discharge. The SEM images of $Li_2O_2$ distribution at the oxygen side, middle part, and separator side and the statistical analysis of particle size at the concentration of (**b**) 0.5 M and (**c**) 2 M. The scale bars in (**b, c**) are 200 nm. **d** The summary of average $Li_2O_2$ diameters with 0.5–2 M electrolytes. The error bars in (**d**) represent the standard deviation derived from numerous particle diameter measurements. The simulated distribution of oxygen concentration at (**e₁**) 60%, (**e₂**) 80%, and (**e₃**) 100% DODs in the 0.5 M electrolyte and at (**f₁**) 60% and (**f₂**) 100% DODs in the 2 M electrolyte. **g** The average

oxygen concentration and the corresponding voltage-capacity curves. The simulated distribution of $Li_2O_2$ volume fraction at (**h₁**) 60%, (**h₂**) 90%, and (**h₃**) 100% DODs in the 0.5 M electrolyte and at (**i₁**) 80% and (**i₂**) 100% DODs in the 2 M electrolyte. The positions of the $Li_2O_2$ maximum volume fractions are marked by white dots. **j** Migration of the $Li_2O_2$ maximum volume fraction site in the electrode depth direction as the discharge proceeds. The simulated distribution of local current discharge along the electrode surface in (**k**) 0.5 M and (**l**) 2 M electrolytes, with $L_{pos} = 0$ representing the oxygen face and $L_{pos} = 1$ representing the separator face. **m** Scheme of mass transport characteristics in 0.5 and 2 M electrolytes.

The walls of the channel are covered with a layer of small particles, as shown in Supplementary Fig. 14 and Fig. 3c.

Figure 3e−g shows the correlation between oxygen concentration and discharge capacity. The oxygen is the rate-limiting step in the electrochemical reaction for its concentration is much lower than the $Li^+$ ion concentration. Higher viscosity reduces the ability to dissolve and transport oxygen, which is the primary factor influencing the discharge capacity and product morphology for the 0.5 M and 2 M electrolytes. For example, at 60% depth of discharge (DOD), the minimum oxygen concentration in the 2 M electrolyte drops to 0.1 mM (Fig. $3f_1$). In comparison, it is 35.8 times higher in the 0.5 M electrolyte (Fig. $3e_1$ and $3f_1$). Combined with the analysis of Sections 2.1 and 2.2, the maximization of discharge capacity in 0.5 M electrolyte can be attributed to the following factors: (*i*) the relatively high ion conductivity, which reduces the ohmic resistance and charge transfer resistance; (*ii*) the nucleation mechanism for particle growth, prevents rapid electrode passivation and further mitigates polarization; (*iii*) the fast oxygen transfer characteristics, allows the electrode pore space to be efficiently utilized.

Notably, the diameters of $Li_2O_2$ particles on the oxygen side, middle part, and separator side are counted. It is commonly assumed that the $Li_2O_2$ distribution is determined by the oxygen concentration gradient, thus more $Li_2O_2$ is thought to appear at the oxygen inlet[16,31]. Interestingly, it is found here that the particle size is distributed inversely to the oxygen gradient in the 0.5 M electrolyte, which exhibits 196, 211, and 239 nm from the oxygen to the separator side (Fig. 3b). While the distribution trend in the 2 M electrolyte follows the oxygen gradient in accordance with the general knowledge (Fig. 3c). The summary of the average size distribution is shown in Fig. 3d and Supplementary Fig. 15. $Li_2O_2$ growing in the 0.05 M and 0.1 M electrolytes is not included in the size statistics because it exhibits a film-like shape.

To explain the anomaly, the evolution of $Li_2O_2$ is further traced considering $Li_2O_2$ particles and possibly $Li_2O_2$ films. According to the $Li_2O_2$ nucleation-growth theory, electrode passivation is contributed by the $Li_2O_2$ film, and the pore space is occupied by $Li_2O_2$ particles. The modeling methodology is detailed in the Supplementary Information. Figure 3j captures the migration of the maximum $Li_2O_2$ volume fraction ($Vf_{max}$) site. In the 0.5 M electrolyte, it can be divided into four stages. Initially, the $Li_2O_2$ $Vf_{max}$ site remains on the oxygen face before 85% DOD (Stage I, Fig. $3h_1$). Subsequently, it starts to migrate toward the separator side (Stage II). During the 90%-97% DOD, it reaches and remains on the separator face (Stage III, Fig. $3h_2$). Finally, it returns toward the oxygen side and is located 8 μm ($L_{pos}$ = 0.84) from the separator face (Stage IV, Fig. $3h_3$). The generation rate and concentration of intermediates $LiO_2$ are directly determined by the local current density, thereby influencing the distribution of $Li_2O_2$. According to the electron tunneling effect, as the thickness of the $Li_2O_2$ film increases, the electrode surface gradually loses activity and finally leads to an extremely slow electrochemical rate (current). Supplementary Fig. 16a indicates that $Li_2O_2$ film is preferentially deposited on the oxygen side and exhibits a gradient similar to oxygen. As a result, at 60% DOD, the peak of local current density starts to move from the oxygen face (50% DOD, Fig. 3k) to the separator side (80% DOD, Fig. 3k). Supplementary Fig. 17 shows the peak migration of local current density. Compared to Fig. 3j, the migration of the $Li_2O_2$ $Vf_{max}$ site exhibits a lag, which is in accordance with the real situation. Then, the sufficient oxygen supply in the 0.5 M electrolyte allows for a sustained migration of the current peak. Until 95% DOD, the peak reaches the farthest site, located 9.5 μm from the separator face ($L_{pos}$ = 0.81). However, the oxygen depletion on the separator face at the end of discharge (Fig. $3e_3$) leads to the peak flowing back to the $L_{pos}$ = 0.77 (100% DOD, Fig. 3k). The above processes are dynamically shown in Supplementary Movie 1. Hence, it is evident that the local deactivation

of the electrochemical surface and sufficient oxygen supply are crucial factors causing the non-conventional distribution of $Li_2O_2$.

Furthermore, Fig. 3i simulates the distribution of $Li_2O_2$ in the 2 M electrolyte, where the $Li_2O_2$ $Vf_{max}$ consistently remains on the oxygen face, consistent with the SEM results. Although the accumulation of $Li_2O_2$ film on the oxygen face (Supplementary Fig. 16b) facilitates the peak of local current density to migrate towards the separator, the low concentration of oxygen there (Fig. 3f) cannot support the electrochemical reactions. Consequently, the peak of local current density eventually reaches 14.5 μm from the oxygen face ($L_{pos}$ = 0.29), and more than half of the electrode is barely utilized due to insufficient oxygen (Fig. $3f_5$), as shown in Fig. 3l. The dynamic processes in the 2 M electrolyte are shown in Supplementary Movie 2. The characteristics of mass transport in 0.5 M and 2 M electrolytes are illustrated in Fig. 3m.

## Impact of local electron transport failures on $Li_2O_2$ growth based on cross-scale modeling

To quantitatively understand the asymmetric growth behavior of $Li_2O_2$ particles in the same channel, a cross-scale model is further developed. Since the oxygen inlet of the channel is easily passivated based on the above 3D heterogeneous modeling, an extreme assumption is made that no oxygen reduction current occurs in this "failed area". The computational domain and reactions are illustrated in Fig. 4a, showing the oxygen can be only electrochemically reduced in the "active area". The solid phase, i.e., $Li_2O_2$ particles, is marked as 1 by the order parameter ($\zeta$). The growth behavior after nucleation is focused, with two identically sized nuclei initially set on the oxygen and separator sides ($L_{pos}$ = 0.125 and 0.875), respectively. The growth rate ($k_g$) of the particle is directly determined by the disproportionation kinetics of the intermediate $LiO_2$ at the $Li_2O_2$/electrolyte interface. The detailed modeling process is described in Methods, and the parameter values used in the simulation are listed in Supplementary Table 5.

The cases of 0%, 10%, and 20% electrochemical area loss ($A_{loss}$) on the oxygen inlet are shown in Fig. 4b. Without any area loss, the $Li_2O_2$ particle on the oxygen side is larger than that on the separator side. As the area loss increases, it is evident that the particle on the oxygen side shrinks while that on the separator side expands. $\zeta$ distributed along the central axis ($L_{channel}$) is shown in Fig. 4c. On the oxygen side, the order parameter of the interface layer ($0 < \zeta < 1$) with smaller area loss is higher than that with larger area loss. However, the situation is reversed on the separator side. The size of $Li_2O_2$ particles on the two sides under the influence of the failed area is summarized in Fig. 4d.

The oxygen concentration and streamlines are shown in Fig. 4e. Compared to 0% loss, the 20% loss results in a higher oxygen concentration at the oxygen inlet, followed by a pronounced concentration drop in the active area (Fig. 4f). According to $\int_{0.2}^1 i_{loc} dL_{active} = I$, this large oxygen gradient is attributed to the higher local current density carried by the 80% electrode area. The peak of local current density is usually located at the boundary between the failed and active area under well-oxygenated conditions, which has also been proven in Fig. 3k, i. The rapid consumption of oxygen between $L_{pos}$ = 0.3–0.8, in the case of 20%, produces more $LiO_2$ in this region (Fig. 4b). With increasing failed area, the $LiO_2$ concentration around the $Li_2O_2$ particles on the oxygen side decreases gradually but increases on the separator side (Fig. 4b), which well explains the changing trend of particle size. Additionally, the $LiO_2$ concentration at the interface of the $Li_2O_2$ particle and electrolyte is lower than that in the electrolyte bulk due to the disproportionation reaction. The evolution of $Li_2O_2$ growth and mass transport are shown in Supplementary Movie 3.

Further, the ratio of $k_g$ (denoted as $r_k$, the growth rate on the oxygen side divided by that on the separator side) at the end of discharge with different area loss is shown in Supplementary Fig. 18. When the area loss is between 0-6%, $r_k$ is greater than 1, meaning that the growth rate on the oxygen side is always larger than that on the

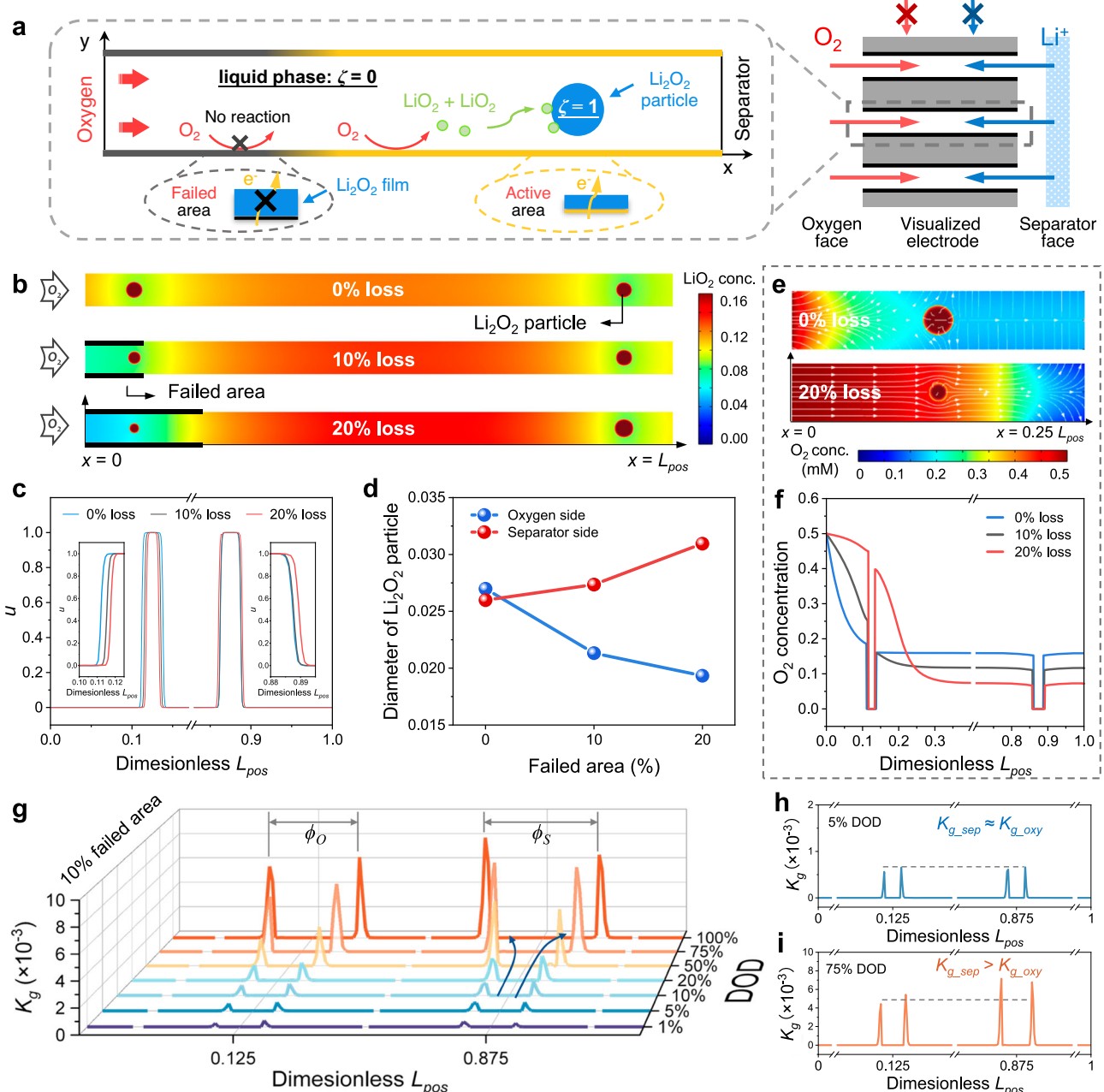

**Fig. 4 | Multi-field cross-scale modeling and mesoscopic analysis of Li$_2$O$_2$ particle growth. a** Scheme of the computational domain and reaction process. **b** Growth results of Li$_2$O$_2$ particles at the oxygen and separator sides and LiO$_2$ concentration distribution in the presence of 0%, 10%, and 20% electrochemical area loss. The dark red circles represent Li$_2$O$_2$ particles. **c** Order parameter along the central axis (i.e., 0.5 $L_{channel}$). **d** The summary of Li$_2$O$_2$ particle diameter at the oxygen and

separator sides in the presence of different electrochemical area loss. **e** The comparison of oxygen concentration at the inlet in the presence of 0 and 20% electrochemical area loss. **f** Oxygen distribution along the central axis. **g** Time-dependent growth rate ($k_g$) of Li$_2$O$_2$ particles along the central axis at 10% electrochemical area loss. The $\Phi_O$ and $\Phi_s$ represent the diameters of Li$_2$O$_2$ particles on the oxygen and separator sides, respectively. The growth rates at DODs of (**h**) 5% and (**i**) 75%.

separator side (Supplementary Fig. 19a,b). However, when the loss exceeds 6%, $r_k$ undergoes a transition (Supplementary Fig. 19c). Taking 10% loss as an example, the time-dependent growth rate of the Li$_2$O$_2$ particle is shown in Fig. 4g and Supplementary Fig. 20. At DODs below 5%, the growth rates of the two particles are almost identical (Fig. 4h). As the discharge proceeds, more aggregation of LiO$_2$ on the separator side leads to a faster increase in the growth rate here. By 75% DOD, the average growth rate on the separator side far exceeds that on the oxygen side, with a ratio of 1.3 (Fig. 4i). Upon further increase to 100% DOD, the ratio becomes 1.78. $r_k$ exhibits a linear decline within the range of 0−30% loss, conforming to $r_k = -3.57\ A_{loss} + 1.23$

(Supplementary Fig. 18). However, for area losses exceeding 30%, $r_k$ experiences an exponential decline.

## Proof of concept and bottleneck breakthrough
The nucleation and transport kinetics, as well as the battery failure mechanisms, are summarized in Fig. 5a. The 0.5 M electrolyte establishes an optimal trade-off between nucleation and transport kinetics, which reverses the Li$_2$O$_2$ distribution and exhibits significant potential for high capacity. High initial nuclei density and consequent Li$_2$O$_2$ film are facilitated in the electrolytes with lower Li$^+$ ion concentration, accelerating the electrode passivation and rapid death. On the other

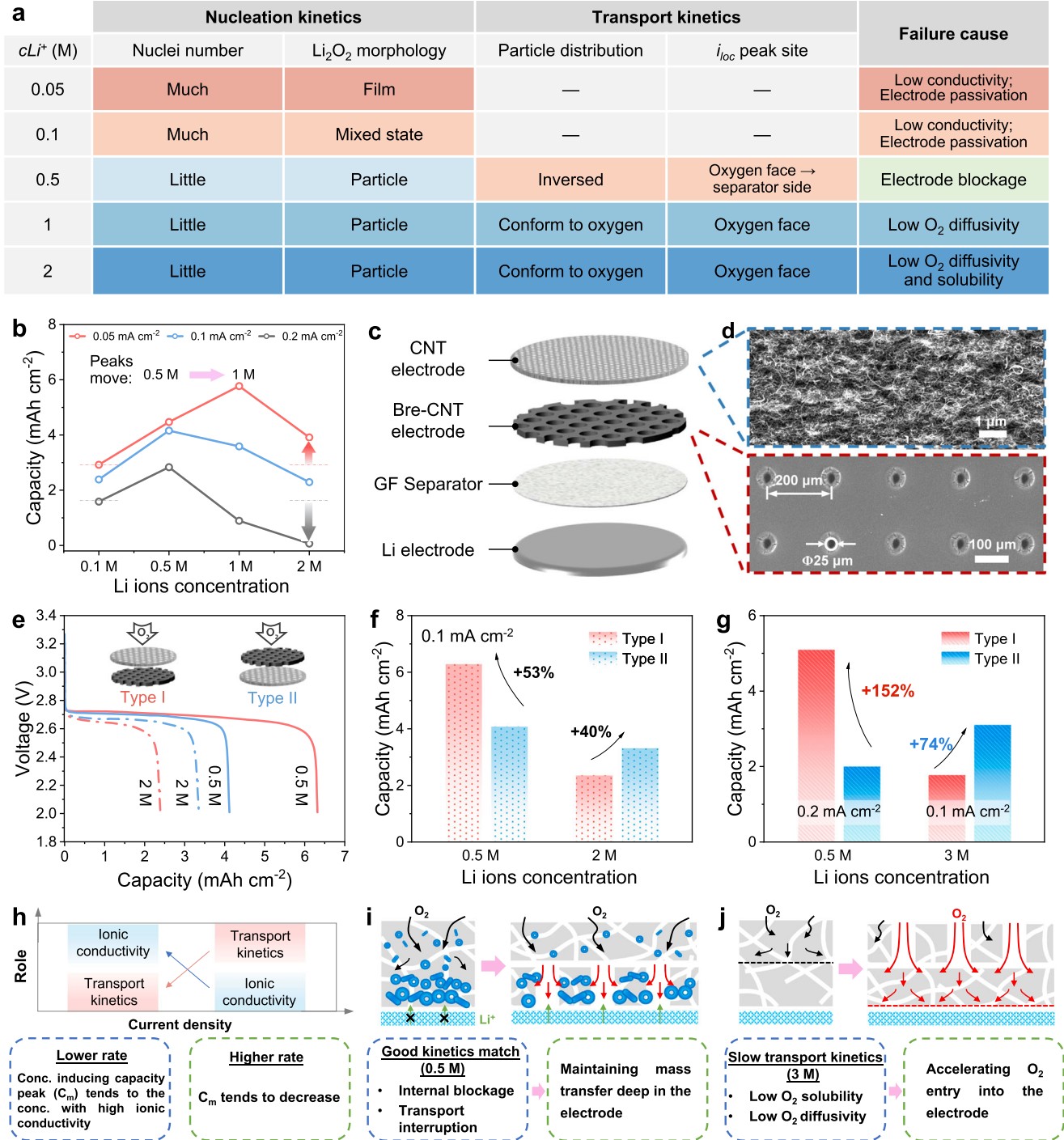

**Fig. 5 | Proof of concept and optimization strategies of general electrodes for high energy density. a** Summary of Li+ ion concentration regulation mechanisms. **b** The capacity trend at different discharge rates. The color gradient indicates the extent of similarity or difference among phenomena or failure causes. **c** Scheme of the double-electrode structure. **d** SEM images of CNT and bre-CNT electrodes. **e** Galvanostatic discharge curves at 0.1 mA cm−2 and (**f**) capacity trend of different structures using 0.5 M and 2 M electrolytes. Red represents structure type I, and blue represents type II. Solid lines represent 0.5 M electrolytes, and dotted lines represent 2 M electrolytes. **g** The capacity trend under amplificative conditions: i) 0.2 mA cm−2 and 0.5 M; ii) 0.1 mA cm−2 and 3 M. **h** Analysis scheme of the impact of discharge rates on capacity peak. The general optimization strategies for the cases with (**i**) a good match between nucleation and transport kinetics and (**j**) slow transport kinetics.

hand, excessive Li+ ion concentrations restrict oxygen transport, resulting in electrode starvation and inefficient utilization.

To prove the concept, three steps are conducted using generally disordered electrodes: (i) tuning the operating protocol (Fig. 5b), (ii) tuning the electrode framework (Fig. 5c–f), and (iii) the amplification experiment (Fig. 5g). Figure 5b shows a shift in Li+ ion concentration inducing the capacity peak from 0.5 M to 1 M when the rates from

0.2 mA cm−2 to 0.05 mA cm−2. As the discharge rate decreases, the capacity of the 2 M electrolyte increases and surpasses that of the 0.1 M electrolyte. This is due to the weakening of the role of transport kinetics at lower rates and the less severe electrode passivation with high Li+ ion concentration. Thus, with lower discharge rates, the capacity peak tends to move towards the Li+ ion concentration with higher ionic conductivity (Fig. 5h).

Inspired by the visualization and quantitative results, the optimization strategy should be tailored based on the degree of alignment with both transport and nucleation kinetics. The electrode framework is tuned to vary transport kinetics. A battery with a double layer of CNT electrodes is assembled, as shown in Fig. 5c. An array of through channels with a central distance of 200 μm and a pore diameter of 50 μm is constructed by lasers on one of the electrodes (Fig. 5d), called the breathing CNT (bre-CNT) electrode. The performance of bre-CNT electrode is shown in Supplementary Fig. 21. Structure type I refers to the stacking of the CNT electrode exposed directly to the oxygen atmosphere on top of the bre-CNT electrode, while structure type II refers to the opposite arrangement (Fig. 5e). The local transport ability of the electrodes is regulated by the position of the breathing channels with low tortuosity, e.g., the transport ability on the separator side of type I is better. In addition, the electrochemical reaction area is equal for both types, providing a good condition for comparison.

In the 0.5 M electrolyte with the optimal kinetics compatibility, Fig. 5e, f shows that type II exhibits a capacity of 4.12 mAh cm$^{-2}$, which is promoted to 6.31 mAh cm$^{-2}$ with type I. The breathing channels in type I maintain the species transport on the severely blocked separator side, and thus the capacity is promoted by 53%. Further, at the double discharge rate with faster oxygen consumption, the capacity is significantly boosted by 152% (Fig. 5g and Supplementary Fig. 22). Clearly, for the electrolytes with decent transport capability, addressing the pore blockage on the separator side is the key to breaking the capacity bottleneck, rather than focusing on oxygen transport (Fig. 5i). However, in the electrolyte with high Li$^+$ ion concentration, the capacity of type II is higher 40% than type I in 2 M electrolyte (Fig. 5e, f), and further higher 74% in 3 M electrolyte with worse transport kinetics (Fig. 5g and Supplementary Fig. 22). The critical factors limiting capacity are low solubility and diffusivity of oxygen. Therefore, the immediate priority for sluggish transport kinetics is to enhance the rapid ingress of oxygen into the electrode (Fig. 5j) by layered or gradient structure designs. For the same electrode materials and electrolyte system, a higher full discharge capacity indicates better maintenance of molecular, ionic, and electronic transport pathways, enhancing both practical capacity and cyclability (Supplementary Fig. 23).

In summary, this work significantly boosts the practical capacity limits by redefining the connection between nanoscale Li$_2$O$_2$ behaviors and macroscopic electrochemical performance. By leveraging the inherent regulatory ability of battery systems, the initial states of nucleation and transport kinetics are controlled. Specifically, a multi-field cross-scale model, combined with visualization techniques, is developed to provide a quantitative and intuitive understanding of the coupling of phase transition and species transport. First, a Li$_2$O$_2$ nucleation-growth theory is proposed. The initial nuclei density and the early voltage are regulated by the reduction kinetics of adsorbed oxygen. High nuclei density tends to form in low Li$^+$ ion concentration electrolytes, resulting in a Li$_2$O$_2$ film that causes a substantial increase in impedance and rapid voltage drop. Importantly, Li$_2$O$_2$ distributed against oxygen gradient in the 0.5 M electrolyte implies the compatibility of nucleation and transport kinetics, thereby achieving maximum capacity. Sufficient oxygen supply and local electron transfer failure are essential for the migration of the current peak and reversed Li$_2$O$_2$ distribution. Furthermore, these findings are successfully proven using general electrodes, underscoring the need to tailor optimization strategies to ensure compatibility with kinetics. For optimal kinetics compatibility, the key to breaking the capacity bottleneck is maintaining the mass transport deep within the electrode, instead of just accelerating oxygen diffusion at the oxygen inlet. As a proof of concept, the capacity limit is boosted by 150% by introducing breathing channels on the separator side. This work overcomes the knowledge limitations, and the revealed mechanism can be extended to other metal-air batteries.

## Methods

### Electrode preparation

For the visualized electrode, the AAO template was selected as the substrate. The AAO membranes with a diameter of 390 nm and a thickness of 50 μm were purchased from Shenzhen Topmembranes Technology Co., Ltd. The precursor solution is prepared by mixing a 15% sucrose solution with ethanol in a 1:1 volume ratio. A simple filtration-calcination method was employed. The AAO membrane was placed on a filtration platform. During the filtration process, 10 ml of pre-prepared solution was dropped onto the AAO membrane, ensuring uniform liquid filtration across the AAO membrane. The solution-treated membrane was transferred into a desiccator at 60 °C for 6 h. Subsequently, the sucrose-treated membrane was calcined in an argon-protected tube furnace at 900 °C for 1 h with a ramp-up and ramp-down rate of 2 °C/min. The carbon load of a visualized electrode was ~ 0.1 mg. For the general disordered electrode, the commercial CNT electrodes with a thickness of 50 μm were purchased from Nanjing XFNANO Materials Tech Co., Ltd. For the bre-CNT electrode, the arrays of breathing channels were punched on the CNT electrode using the Laser ultra-precision processing system (UP-D).

### Battery assembly and electrochemical methods

The batteries were assembled in an Ar-filled glove box (Etelux, Lab 2000). The H$_2$O and O$_2$ contents were below 0.1 ppm. A homemade Swagelok-type battery was used. For the battery with a single-air electrode structure, a commercial Li foil, a separator (Whatman GF/C 1822), a CNT or visualized electrode, and a stainless steel mesh for the current collection were stacked sequentially, with 100 μL electrolyte added. 0.05, 0.1, 0.5, 1, 2, and 3 M LiTFSI (99.9%, Sigma)/TEGDME (Suzhou Duoduo Chemical Technology Co., Ltd.) electrolytes were self-configured. For the double-electrode structure, a CNT electrode and a bre-CNT electrode were tightly fitted together and replaced the previous single electrode. 200 μL electrolyte was added to ensure electrode wetting. The assembled battery was purged with oxygen (purity > 99.999%) to remove argon and allowed to stand for 1 hour. The oxygen valve was then closed, and the battery was left to stand for an additional 2 hours. The galvanostatic discharge was tested using the Neware (CT3008W) with a cut-off voltage of 2.0 V versus Li/Li$^+$. All electrochemical tests are conducted at room temperature.

### Characterizations

After discharge, the batteries were dissembled in the glove box for the following characterizations. The discharged electrodes were rinsed with 1,2-Dimethoxyethane (DME) and dried. Li$_2$O$_2$ distribution and morphology were observed using SEM (JEOL-7500 F) at an accelerating voltage of 3.0 kV. Li$_2$O$_2$ composition and crystal type were detected by XRD (miniflex600). $C_{dl}$ was tested using the Cyclic Voltammetry (CV) method with the voltage region of 2.90–2.96 V at the 0.2–1.0 mV s$^{-1}$ scan rates.

### Multi-physics cross-scale model

The microscale growth of Li$_2$O$_2$ particles in the channel was described by the phase field method. The macroscopic continuum method was used to simulate the mass and electron transport in the electrolyte and on the electrode surface. The two scale methods were connected by an important parameter, the LiO$_2$ concentration term, transported at the interface of Li$_2$O$_2$ particle and electrolyte (solid-liquid interface). The governing equations are detailed in the Supplementary Information. The calculations are conducted using COMSOL Multiphysics 6.0.

## Data availability

The data supporting the findings of this study have been deposited in the Figshare under the accession code https://doi.org/10.6084/m9.figshare.27199287.

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

## Acknowledgements

The authors thank the funding support from the National Natural Science Foundation of China (52376080 and 52306122), Anhui Provincial Natural

Science Foundation (2308085QE174), China Postdoctoral Science Foundation (2023TQ0346 and 2024M753112), Postdoctoral Fellowship Program of CPSF (GZC20232522), Fundamental Research Funds for the Central Universities (WK2090000057), and Students' innovation and Entrepreneurship Foundation of USTC (CY2023C008).

## Author contributions

Z.Z., X.X., and P.T. designed the experiment; Z.Z. performed research; Z.Z., A.Y., K.S., and J.Y. analyzed data; Z.Z. wrote the paper; X.X. and P.T. wrote the review & editing, provided supervision, project administration, and funding acquisition.

## Competing interests

The authors declare no competing interest.
