## [Transparent Peer Review file · Nature Communications]

Breaking the Capacity Bottleneck of Lithium-Oxygen Batteries through Reconceptualizing Transport and Nucleation Kinetics

Corresponding Author: Professor Peng Tan

Version 0:

Reviewer comments:

Reviewer #1

(Remarks to the Author)

This manuscript by Zhang et al. presents combined experimental and modeling study to redefine the relationship among mass transports, Li₂O₂ nucleation/growth and discharge properties of Li-O₂ battery. Through a systematic control of electrolyte concentration, the authors found that Li₂O₂ particle distributed against oxygen gradient is the result of a trade-off between nucleation and transport kinetics. In addition, they suggested a double-stacked architecture of air electrodes that can promote mass transport and thus enhance the discharge capacity about 150%. Experimental methodology using carbon coated AAO electrode is quite impressive and data interpretation is clear. However, the evidence on the main insistence mostly rely on multi-physics simulations with a simple experimental observation. Basically, the findings do not extend beyond current understanding on the factors (i. surface passivation by Li₂O₂ film formation, ii. pore clogging by Li₂O₂ particle with limited oxygen transport) that limit a true discharge capacity of Li-O₂ battery in the community. Therefore, this paper does not meet the high standard of Nature Communication at least for current manuscript. The following issues need being addressed before publication.

1. There are only two data points for disordered electrode in Figure 1. It does not make sense to discuss the tendency with two data point. Same or similar data points to that in visualized electrode may be needed to clarify and guarantee the universality of the findings. Overall, it is concerned that most experimental findings here are a case result of the specific model based on aligned interconnected electrode.
2. Previous studies (DOI: 10.1149/2.0711613jes, 10.1021/acs.jpcc.6b01474) have also shown the effect of electrolyte concentration in Li-O₂ battery even with same salt and solvent. However, the order of discharge capacity is different from this manuscript. What makes this difference? Current density is one of the factors affecting the order of discharge capacity. How did the author determine the current density for the experiments in this work? Applied current density of disordered electrode (0.1 mA cm⁻²) and visualized electrode (300 mA g⁻¹ ~ 0.06 mA cm⁻² based on loading level information) is also different. Rigorous discussions with various experimental evidence may be needed for clarifying current density effects.
3. How do you define initial voltage in Figure 1? Quantitative definition of initial voltage is unclear. What does it mean initial voltage?
4. Li₂O₂ film formation is argued as one of the origins for premature failure in low cLi⁺. Is the finding universal for other electrolyte systems such as high donor-number electrolytes? How could be the mechanism differed in electrolytes containing redox mediator?
5. What is the minor XRD peaks at 43 and 53 degree in Figure S3? If byproduct generated, please also provide the Li₂O₂ formation yield in the experimental setup.
6. How much was discharge capacity obtained for electrode in Fig. 2a-d? Were the SEM images of electrode taken from fully discharged state? As the morphology of Li₂O₂ is also a function of discharge capacity, it is important to elaborate the discharge state or capacity for a fare comparison.

Reviewer #2

(Remarks to the Author)

In this work, the authors employ a combination of Multiphysics simulations and experimental characterization in gas diffusion

electrodes with a well-controlled, AAO-based microstructure. In doing so they resolve the impact of Li⁺ and O₂ transport on Li_xO₂ morphology/distribution and the resulting electrochemical performance of Li-O₂ batteries. The authors propose that, by adjusting the Li⁺ concentration in the electrolyte, the maximum allowable discharge capacity of the oxygen cathode can be due to either interfacial passivation due to an undesirable LiO₂ morphology, or poor reactant transport either due to pore blockage or low oxygen diffusivity/solubility. The experimental design seems to be well thought-out, and the implementation of AAO as a support to image deposition at various length scales is clever. I recommend publication of the manuscript after the following revisions are considered/addressed:

- In some cases, the clarity of the work would benefit from minor language editing. For example, the meaning of terms like “extreme point” and “tendency” (which I believe correspond to “global maxima” and “trend”?) are relatively difficult to parse in the contexts of which they are used. Unfortunately, I find that this does negatively impact my ability to understand the main conclusions of the work, which are well-communicated in the figures but less so in the text.
- Why are units of areal capacity and specific capacity used when comparing the disordered and AAO air electrodes? This seems relatively counter-intuitive and a rationale should be provided.
- When comparing the impedance behavior of the 1 M and 0.1 M electrolytes before and after discharge, the net change in R_s and R_{ct} at the same capacity (i.e. the same mass of Li₂O₂ formed) should also be investigated to ensure that the differences are actually a result of the electrolyte concentration.
- The limiting current measurements for Li⁺ and O₂ cited by the authors are measured by glassy carbon RDEs and their relevance for gas diffusion electrodes are unclear. Furthermore, the values cited are very similar in magnitude and would be heavily dependent on the electrolyte chemistry (i.e. Li⁺ concentration and O₂ solubility).
- The nucleation-growth mechanism proposed by the authors seems to be more related to growth and has little to do with the nucleation regime, which should encompass the factors associated with initial Li₂O₂ seed formation and coverage before the transport limited regime. It is recommended that this regime either be discussed or the terminology be changed to something more akin to growth mechanism.
- To confirm the author’s proposed passivation film failure mechanism for the low-Li⁺ concentration system, it would be useful for to provide data where the cell is fully discharged, rested for sufficient time to regain reactant concentration at the interface, and then discharged again. If a passivating film is responsible for failure, one would expect there to be negligible capacity output after the rest step due to high interface impedance.
- The values for Li⁺ and O₂ solubility and transport (e.g. diffusion coefficient) used in the electrolyte portion of the Multiphysics simulations should be explicitly stated, discussed, compared to the actual experimental systems, and rationalized on this basis in the main text.
- Similarly, I cannot locate the simulation parameters for the 2 M Li⁺ simulations. These must also be stated, and whether the aforementioned transport parameters were adjusted based on said concentration must be discussed.

Reviewer #3

(Remarks to the Author)

This manuscript provides the correlation between nucleation and kinetic of the Li₂O₂ formation utilizing visualization methods such as C-AAO electrodes and computational modeling. Such an approach may interest researchers to understand the kinetical and structural challenges the LOBs have. However, the information provided here is based on a very limited situation and cannot represent the general phenomena happening in LOBs. Therefore, I suggest rejecting the publication and some questions for manuscript modification.

1) Fig 1A and B show the discharge capacity with each electrode but they have different dimensions, which makes it hard to compare. Moreover, how can the phenomenon shown with a visualized electrode represent that shown with a disordered electrode?

2) Fig 1H only provides the EIS results of the visualized electrode before discharge even rest of manuscript are based on the results utilizing C-AAO electrode. Results shown after discharge with C-AAO electrode should be attached.

3) Fig 3E and F show the oxygen flux with different concentrations of electrolyte. After discharge, it seems the electrode filled with particles more densely when 0.5 M was used but oxygen flux was much lower in 2 M. Explanation combined with SEM images should be provided.

4) In Fig 5D, the discharge capacity in each electrode configuration. The electrolyte concentrations are 0.5 M and 3 M, which makes the unity of the overall manuscript poor. Results with 2 M electrolyte should be contained.

5) Data in Fig 5D need to be compared with the cell assembled with single CNT and bre-CNT.

Version 1:

Reviewer comments:

Reviewer #1

(Remarks to the Author)

The manuscript has been well revised with all issues raised by the reviewers properly addressed. I would like to recommend

it for publication after addressing following minor suggestion.

1. It is well known trade-off between capacity and cyclability in most of batteries particularly for lithium oxygen system. It is recommended for the authors to provide recharging properties of your cells that are redesigned to boost practical capacity and suggest some relevant discussions, which might help guide future research insights in this community.

Reviewer #2

(Remarks to the Author)

I believe that the authors have sufficiently rectified the outstanding issues noted in the first round of review, and therefore recommend publication.

Reviewer #3

(Remarks to the Author)

All questions were answered well enough. However, most of the findings in this work are not the next of currently established knowledge we have in LOBs. (e.g. oxygen gradient, salt concentration, or current density related to the morphology or formation of discharge products) The authors utilized C-AAO to observe the gas diffusion layer in cross-section, emphasizing it as a novel methodology, which has already been reported. Therefore, this work does not have enough novel points required for publishing Nature Communication. Belows are questions still remains.

- 1) The difference in applied current density in CNT and C-AAO will affect the electrolyte concentration gradient near the electrode surface during discharge. The discussion and additional data should be included, particularly discharge capacity.
- 2) Morphological differences when CNT and C-AAO are discharged at the same current density, which is extending question of 1).

Version 2:

Reviewer comments:

Reviewer #3

(Remarks to the Author)

The authors have responded to all questions properly and this work has done systematically reassessed compared to previous works. Therefore, I recommend the publication to the nature communication.

However, since what was mainly shown in this manuscript was a type of model electrode called AAO, it is not positive in terms of whether it is really useful for lithium-air batteries, which are actually being studied. I hope the author will conduct follow-up research on this in the future.

Response to the First Reviewer's Comments

This manuscript by Zhang et al. presents combined experimental and modeling study to redefine the relationship among mass transports, Li_2O_2 nucleation/growth and discharge properties of Li- O_2 battery. Through a systematic control of electrolyte concentration, the authors found that Li_2O_2 particle distributed against oxygen gradient is the result of a trade-off between nucleation and transport kinetics. In addition, they suggested a double-stacked architecture of air electrodes that can promote mass transport and thus enhance the discharge capacity about 150%. Experimental methodology using carbon coated AAO electrode is quite impressive and data interpretation is clear. However, the evidence on the main insistence mostly rely on multi-physics simulations with a simple experimental observation. Basically, the findings do not extend beyond current understanding on the factors (i. surface passivation by Li_2O_2 film formation, ii. pore clogging by Li_2O_2 particle with limited oxygen transport) that limit a true discharge capacity of Li- O_2 battery in the community. Therefore, this paper does not meet the high standard of Nature Communication at least for current manuscript. The following issues need being addressed before publication.

Comment #1: *There are only two data points for disordered electrode in Fig. 1. It does not make sense to discuss the tendency with two data point. Same or similar data points to that in visualized electrode may be needed to clarify and guarantee the universality of the findings. Overall, it is concerned that most experimental findings here are a case result of the specific model based on aligned interconnected electrode.*

Response: We thank the reviewer for the important suggestion. The use of two types of electrodes in Fig. 1 aims to validate the consistency of electrochemical behaviors between the visualized and disordered electrodes, thereby confirming the validity and generality of conclusions based on visualized electrodes. Following this comment, revisions have been made in the manuscript:

(i) More data points of disordered electrodes (0.05 M, 0.5 M, and 2 M) are supplemented, consistent with the points of visualized electrodes. The voltage-capacity curves and initial voltage plateau of disordered electrodes with Li^+ ion concentration of 0.05-2 M are shown in **Fig. R1a and Fig. R1d**, respectively. The quantitative definition of the initial voltage plateau is given in Comment #3 (Page R6). **Fig. R1c and Fig. R1f** suggest that the trend of capacity and voltage of the two types of electrodes are the same. The maximum capacity is achieved in 0.5 M electrolyte, and 0.05 M electrolyte leads to the highest initial voltage plateau. Therefore, the electrochemical behaviors of the two types of electrodes are consistent, making it feasible to study nucleation and transport behaviors in lithium-oxygen batteries using the convenient visualized electrodes as tools for disordered gas electrodes.

(ii) Comparison under identical conditions is enhanced. The EIS results at a fixed capacity with various Li^+ ion concentrations are supplemented, as shown in **Fig. R1g-i** and **Table R1**. The net increase in impedance equals to the impedance after discharge minus the pristine value ($R_{net} = R_{discharged} - R_{pristine}$). The criteria for selecting the fixed

capacity is to do so before the rapid voltage decay occurs. For both disordered and visualized electrodes, the highest net R_s and net R_{ct} occur in the 0.1 M electrolyte, significantly surpassing those in electrolytes with high Li^+ ion concentrations. Therefore, the impedance behaviors of the two types of electrodes are also consistent, which further implies the predominant contribution of Li_2O_2 film to the rapid failure of batteries using electrolytes with low Li^+ ion concentrations (Page 6, lines 12-16).

(iii) The purpose of testing with two types of electrodes has been reclarified in the main text: “To ensure the universality of the conclusion, the electrochemical behaviors of disordered (a general carbon nanotube electrode) and visualized electrodes are first compared” (Page 5, lines 12-13) and “The above findings demonstrate a remarkable consistency in both electrochemical and Li_2O_2 behaviors” (Page 7, lines 3-5). Thus, it is believed that the visualized electrodes can reflect the phenomena inside the disordered electrodes”.

(iv) The universality of the conclusions drawn from the visualized electrode is further validated using disordered electrodes in Fig. 5. The proof of concept section has been reorganized and divided into three steps (Pages 13-14): a) tuning the operating protocol, b) tuning the electrode framework, and c) the amplification experiment. In this section, the discussions on the impact of current on the capacity trend with various Li^+ ion concentrations (Fig. 5b and h), the performance enhancement effects of optimization strategies under different degrees of kinetic matching (Fig. 5e and f), and the performance of single CNT and bre-CNT electrodes (Fig. S21) are supplemented.

Fig. 1. Electrochemical behaviors under different Li⁺ ion concentrations. Galvanostatic discharge curves with 0.05-2 M electrolytes using (a) disordered electrodes at 0.1 mA cm⁻² and (b) visualized electrodes at 300 mA g⁻¹. (c) The trend of discharge capacity with Li⁺ ion concentration. Initial voltage plateau using (d) disordered electrodes and (e) visualized electrodes. (f) Trend of initial voltage plateau with Li⁺ ion concentration. EIS results using (g) disordered electrodes at a fixed capacity of 1.5 mAh cm⁻² and (h) visualized electrodes at a fixed capacity of 4000 mAh g⁻¹. (i) Comparison of the net R_s and net R_{ct} , where net R is defined as $R_{discharged} - R_{pristine}$.

Table R1. The values of R_{ct} and R_s of CNT and C-AAO electrodes at a fixed capacity

Electrode type	State	Symbol (Ohm)	0.1 M	0.5 M	1 M	2 M
CNT electrode	Pristine	R_{ct}	226.4	192.9	184.8	88.13
		R_s	92.89	26.80	17.03	26.55
	Discharged	R_{ct}	350.4	257.8	234.8	134.1
		R_s	121.7	27.59	17.45	29.98
	Net	R_{ct}	124.0	64.90	50.00	45.97
		R_s	28.57	0.790	0.420	3.430
C-AAO electrode	Pristine	R_{ct}	743.1	525.4	556.0	450.5
		R_s	128.1	19.68	21.93	19.80
	Discharged	R_{ct}	1086	611.1	630.6	535.5
		R_s	150.3	21.27	22.16	20.81
	Net	R_{ct}	342.9	85.70	74.60	85.00
		R_s	22.20	1.590	0.230	1.010
Equivalent Circuit						

Comment #2: Previous studies (DOI: 10.1149/2.0711613jes, 10.1021/acs.jpcc.6b01474) have also shown the effect of electrolyte concentration in Li-O₂ battery even with same salt and solvent. However, the order of discharge capacity is different from this manuscript. What makes this difference? Current density is one of the factors affecting the order of discharge capacity. How did the author determine the current density for the experiments in this work? Applied current density of disordered electrode (0.1 mA cm⁻²) and visualized electrode (300 mA g⁻¹ ~ 0.06 mA cm⁻² based on loading level information) is also different. Rigorous discussions with various experimental evidence may be needed for clarifying current density effects.

Response: This is a great point, and we thank the reviewer for bringing this up. Firstly, we will clarify the selection criteria for applied current density for the different electrodes. The selection of the current density is a well-considered decision; however, this process was inadvertently omitted during manuscript editing, and we sincerely apologize for this inconvenience.

Due to the substantial structural differences between visualized electrodes (C-AAO electrodes) and disordered electrodes (CNT electrodes), it is deemed unfair to

benchmark them solely based on areal or specific current. To address this disparity, we recalibrate the applied current for the two electrodes using the actual electrochemical active area (A_e) as a reference. Although the total surface area of the electrode ($A_{total} = A_e + A_n$) can be determined through the Brunauer-Emmett-Teller (BET) method, it encompasses areas that do not actively partake in the electrochemical processes (A_n). Therefore, a double-layer capacitance (C_{dl}) method is used. Cyclic Voltammetry (CV) curves of the different electrodes in the Ar atmosphere were measured, as shown in **Fig. R2a-b**. The voltage region is 2.90-2.96 V and the scan rates are 0.2-1.0 mV s^{-1} . C_{dl} is obtained by fitting a function of response current density and scan rate, as shown in **Fig. R2c-d**. Since the C_{dl} value is linearly proportional to the electrochemically active surface area of the electrode, the C_{dl} ratio of the C-AAO electrode and CNT electrode corresponds to their ratio of electrochemical active areas. As a result, $C_{dl, CNT}/C_{dl, AAO} = 22.5 \text{ mF cm}^{-2}/5.07 \text{ mF cm}^{-2} = 4.44$. For the C-AAO electrode, the specific current is $i_{specific, AAO} = 300 \text{ mA g}^{-1}$ and the areal current is $i_{areal, AAO} = 300 \text{ mA g}^{-1} \times 0.1 \text{ mg}/(1.3 \text{ cm}^2)/3.14 \times 4 = 0.0226 \text{ mA cm}^{-2}$. According to the C_{dl} ratio of the two electrodes, the areal current of the CNT electrode is $i_{areal, CNT} = 0.0226 \text{ mA cm}^{-2} \times 4.44 = 0.100 \text{ mA cm}^{-2}$. The parameters of the two electrodes' structure and applied current are summarized in **Table R2** for reader comprehension. The selection of operating protocol has been included in the revised manuscript (Page 5, lines 14-18) and revised Supplementary Information (Fig. S2 and Table S1).

Fig. R2. The double-layer capacitance (C_{dl}) of the air electrode measured by a CV method. CV curves of (a) C-AAO electrode and (b) CNT electrode in the region of 2.90-2.96 V vs. Li at 0.2-1.0 mV s^{-1} scan rates. C_{dl} of (c) C-AAO electrode and (d) CNT electrode.

Table R2. The parameters of C-AAO electrode and CNT electrode

Parameters	C-AAO electrode	CNT electrode
Diameter	13 mm	8 mm
C_{dl}	5.07 mF cm ⁻²	22.5 mF cm ⁻²
Applied current density	0.0226 mA cm ⁻² (300 mA g ⁻¹)	0.1 mA cm ⁻² (20 mA g ⁻¹)

The references provided by the reviewer hold valuable insights. The first reference used LiTFSI/TEGDME, which is the same as ours. The second reference used LiTFSI/DME electrolyte. Their conclusions do not conflict with ours and are even consistent. In the first reference, the differences between capacities under various Li⁺ ion concentrations are very small (the capacities of 0.5 M and 1 M electrolytes are almost the same at 0.1 and 0.2 mA cm⁻²), making it difficult to discern precise capacity trends. According to the results of this reference, we can only know that the optimal concentration is in the range of 0.5-1 M, aligning closely with our findings. Notably, the optimal concentration is determined by the current. Since this reference does not provide the current density based on the actual electrochemical active area, the actual current may significantly differ from our experiments. To clarify the effect of the current, we further discuss it and supplement additional experiments.

The capacity trend affected by the current densities can be inferred based on our theory. The rate of oxygen consumption certainly increases at high current densities. In this case, the electrolyte's ability to transport oxygen will play a more crucial role. Electrolytes with less Li salt have higher solubility and diffusivity of oxygen. Therefore, with the increase in current, the Li⁺ ion concentration inducing capacity peak tends to decrease. On the contrary, with the decreasing currents, the role of transport kinetics will be weakened, while the ionic conductivity will be relatively elevated. In this case, the capacity peak tends to move towards the Li⁺ ion concentration with higher ionic conductivity, as illustrated in **Fig. R3a**. The capacity trends under current densities 0.05, 0.1, and 0.2 mA cm⁻² are supplemented in **Fig. R3b**. This result is consistent with our speculation and theory. The sampling interval of Li⁺ ion concentration may be wide, resulting in the capacity peak under 0.1 and 0.2 mA cm⁻² both remaining at 0.5 M in Fig. R3b. The effect of current density well proves our theory and has been included in the proof of concept section in the **revised manuscript (Fig. 5b and h, and lines 1-8 on page 14)**.

From the results, our theory can be also extended to other solvent systems, such as DME used in the second reference. In this reference, with the decrease in current density, the capacity of a 2 M electrolyte also increases from below that of the 0.1 M electrolyte to above it. This is highly consistent with our theory and results.

Fig. R3. (a) Analysis scheme of the impact of discharge rates on capacity peak based on the theory proposed in this work. (b) The capacity trend with the disordered electrode under 0.05, 0.1, and 0.2 mA cm⁻².

Comment #3: How do you define initial voltage in Fig. 1? Quantitative definition of initial voltage is unclear. What does it mean initial voltage?

Response: We are sorry for this unclear definition. We will elucidate the purpose of defining this variable and provide its quantitative definition.

To demonstrate the formation of Li₂O₂ film in low-concentration electrolytes, we trace back to the nucleation of Li₂O₂. In the early stage of discharge, LOBs undergo a rapid voltage drop followed by a stable voltage plateau. During this voltage drop, the electrochemical reduction of oxygen first occurs (O₂ + e⁻ → O₂⁻) rather than direct phase transition (Li₂O₂ deposition). As emphasized in the manuscript, this process should be distinguished from metal deposition. Further, we stated that “The decrease in salt concentration enhances the oxygen solubility^{1,2}, while simultaneously freeing up sites on the electrode surface for oxygen molecules to absorb before discharge (Fig. 2e). Therefore, a negative correlation between the initial voltage plateau and Li⁺ ion concentration in Fig. 1d-f is observed (Fig. 2f)”. The reduction rate of adsorbed oxygen is crucial for the initial nuclei density of Li₂O₂. Thus, the “initial voltage” can reflect the oxygen reduction kinetics before the Li₂O₂ nucleation.

Based on the statements above, it can be seen as a turning point in the early discharge, i.e., the beginning of the voltage plateau. For ease of understanding, we have amended the term “initial voltage” to “initial voltage plateau”. To accurately capture this inflection point, the dV/dQ method is employed, which is a commonly used approach to identify the change in voltage. The dV/dQ plots are shown in **Figs. R4 and R5**, where the peak corresponds to the “initial voltage plateau”. The initial voltage plateau decreases with an increase in Li⁺ ion concentration, indicating lower oxygen reduction kinetics and less nucleation in the early discharge of high-concentration electrolytes. The definition of initial voltage plateau has been included in the **revised manuscript (Page 5, lines 27-28)** and **revised Supplementary Information (Figs. S3 and S4)**.

Fig. R4. Discharge curves and dV/dQ plots of disordered electrodes with Li^+ ion concentration of (a) 0.05 M, (b) 0.1 M, (c) 0.5 M, (d) 1 M, and (e) 2 M. (f) The trend of initial voltage plateau with varying Li^+ ion concentrations.

Fig. R5. Discharge curves and dV/dQ plots of visualized electrodes with Li^+ ion concentration of (a) 0.05 M, (b) 0.1 M, (c) 0.5 M, (d) 1 M, and (e) 2 M. (f) The trend of initial voltage plateau with varying Li^+ ion concentrations.

Comment #4: Li_2O_2 film formation is argued as one of the origins for premature failure in low $c\text{Li}^+$. Is the finding universal for other electrolyte systems such as high donor-number electrolytes? How could be the mechanism differed in electrolytes containing redox mediator?

Response: We thank the reviewer for this good comment. The donor number (DN) of the solvent affects the adsorption/dissolution state of O_2^- , thereby determining the morphology and properties of the products. We use dimethyl sulfoxide (DMSO) with a high DN as the solvent and LiTFSI as the lithium salt. For a fair and direct comparison,

all operating conditions of DMSO experiments are the same as the TEGDME experiments. The current density is 0.1 mA cm^{-2} and the fixed capacity is set to be 1.5 mAh cm^{-2} . The disordered electrodes with 0.1 M and 1 M LiTFSI/DMSO are tested, respectively. The SEM images of product morphologies are shown in **Fig. R6**. In the 0.1 M electrolyte, the Li_2O_2 shows a film-like structure, covering the electrode surface. While in the 1 M electrolyte, Li_2O_2 is typically toroidal with the exposed active surface of the electrode. Correspondingly, the R_{ct} of 0.1 M electrolyte is notably higher than that of 1 M electrolyte, as shown in **Fig. R7**. Both the product morphologies and the EIS trends of the DMSO experiments are similar to TEGDME experiments. Therefore, we think that “ Li_2O_2 film is one of the origins for premature failure in low $c\text{Li}^+$ ” can be preliminarily extended to high-DN electrolyte systems. The above results and discussion have been included in the **revised manuscript (Page 8, lines 22-25)** and **revised Supplementary Information (Fig. S10)**

Fig. R6. The SEM images of Li_2O_2 morphologies on the disordered electrodes with a fixed capacity of 1.5 mAh cm^{-2} in (a) 0.1 M and (b) 1 M LiTFSI/DMSO.

Fig. R7. The EIS results at a fixed capacity of 1.5 mA cm^{-2} in (a) 0.1 M and (b) 1 M LiTFSI/DMSO.

Redox mediators (RMs) can generally be classified into those used for discharge (D-RMs) and for charge (C-RMs), and this work focuses on the discharge processes. The electrochemical processes of RMs and their chemical interactions with active species are complex, which depend on the nature of the RMs. For example, ethyl viologen (EtV^{2+}) and 2,5-di-tert-butyl-1,4-benzoquinone (DBBQ) have different processes. During discharge, $\text{EtV}^{2+}/\text{EtV}^+$ acts as the sole intermediate, solely serving as an electron shuttle (Reactions R1-R4)³. For the DBBQ, a new intermediate, LiDBBQ, is formed and directly participates in forming solid Li_2O_2 (Reactions R5-R8)⁴. The

failure mechanism of electrolyte systems containing RMs will be a new theoretical framework, potentially varying with the type of RMs, which may extend beyond the scope of this work. Furthermore, the effects and mechanisms of D-RMs at different Li^+ ion concentrations have been overlooked. We are pleased to discuss this issue with the reviewer and provide our speculations based on the conclusions of this manuscript.

Since D-RMs promote the solubility of intermediates, the electrode can maintain the active area. We think that for the D-RMs, the role of transport kinetics will be promoted, while the electron transport limitation will be weakened to some extent. Our hypothesis is as follows: In the electrolytes with low Li^+ ion concentrations (0.05 M or 0.1 M) and the presence of D-RMs, Li_2O_2 tends to shift from film to particle. Conversely, in the electrolyte with high Li^+ ion concentrations (2 M or 3 M), transport kinetics, especially oxygen transport, become the predominant factor. In this case, the role of D-RMs in enhancing capacity may be less significant, as D-RMs primarily improve electron transfer. It has been reported that the addition of RM reduces the oxygen solubility in the electrolyte⁵, potentially worsening oxygen transport. Therefore, compared to RM-free electrolytes, the presence of D-RMs tends to lower the Li^+ ion concentration at which the capacity peak occurs. This is an interesting and hot topic, which may be covered in our future work. We hope the reviewers will remain engaged with our upcoming works.

For a perspective, (i) by utilizing the visualized electrode proposed by this work, Li_2O_2 distribution affected by the D-RMs inside the gas electrode can be further explored, thus potentially elucidating the optimal electrode design suitable for D-RMs. (ii) By-products due to RMs instability may also be the cause of failure. Most studies focus on the performance of RMs while overlooking their stability assessment. (iii) The kinetic parameters (e.g., solubility and diffusivity of species) under various conditions (e.g., concentrations of Li^+ ion and RMs) are lacking in research, yet it is crucial for understanding the failure mechanism of electrolyte systems containing RMs.

Comment #5: *What is the minor XRD peaks at 43 and 53 degree in Fig. S3? If byproduct generated, please also provide the Li_2O_2 formation yield in the experimental setup.*

Response: We are sorry for this inconvenience caused by the lack of clear labeling. The XRD peaks at 43° and 53° in the original figure are the signals of the CNT electrode, not byproducts. The XRD patterns in the pristine manuscript are from the fully

discharged state. The signals of the CNT electrode around 37.5°, 43°, and 53° are weakened due to electrode coverage by the discharge product.

To elucidate this phenomenon, the XRD results of the pristine electrode and the discharged electrodes at a fixed capacity of 1.5 mAh cm⁻² are further tested. **Fig. R8** demonstrates the identification of the product as Li₂O₂, indicated by blue shading, with no byproduct present. The signals of the CNT electrode are shaded in gray. The signals at 43° and 53° show a decrease in signal intensity of the CNT electrode due to the coverage of Li₂O₂. The signals of the CNT electrode are weakened more by film-like Li₂O₂ in 0.1 M electrolyte. This result is included in the **revised Supplementary Information (Fig. S7)**.

Fig. R8. The XRD patterns of the pristine CNT electrode and the CNT electrodes discharged to a fixed capacity of 1.5 mAh cm⁻² using 0.1 M and 1 M electrolytes.

Comment #6: *How much was discharge capacity obtained for electrode in Fig. 2a-d? Were the SEM images of electrode taken from fully discharged state? As the morphology of Li₂O₂ is also a function of discharge capacity, it is important to elaborate the discharge state or capacity for a fair comparison.*

Response: We thank the reviewer for pointing out the missing capacity for SEM images in Fig. 2a-d, and we are sorry for this unclear labeling. The SEM images in Fig. 2a-d are obtained from a fully discharged state and correspond to the discharge curves in Fig. 1b with a cut-off voltage of 2.0 V. In the revised manuscript, we have clarified the discharge state of SEM images in Fig. 2a-d.

Furthermore, to fairly compare the inherent regulatory ability of Li⁺ ion concentration, product morphologies at a fixed discharge capacity are supplemented for visualized electrodes (4000 mAh g⁻¹) and disordered electrodes at (1.5 mAh cm⁻²), as shown in **Fig. R9** and **Fig. R10**, respectively. The criteria for selecting the capacity is to do so before the rapid voltage decay occurs. In the electrolytes with low Li⁺ ion concentrations (0.05 M and 0.1 M), Li₂O₂ appears as a film enveloping the surface of CNTs; while it presents as particles with higher Li⁺ ion concentrations (0.5-2 M). The SEM images are supplemented in the **revised Supplementary Information (Fig. S6 and Fig. S8)**.

Fig. R9. The SEM images of Li_2O_2 morphology on the visualized electrode at the fixed capacity of 4000 mAh g^{-1} using (a) 0.05 M, (b) 0.1 M, (c) 0.5 M, and (d) 2 M electrolytes.

Fig. R10. The SEM images of Li_2O_2 morphology on the disordered electrode at the fixed capacity of 1.5 mAh cm^{-2} using (a) 0.1 M, (b) 0.5 M, (c) 1 M, and (d) 2 M electrolytes.

We thank the reviewer for the valuable comments, which are helpful for improving the quality of this manuscript.

Response to the Second Reviewer's Comments

In this work, the authors employ a combination of Multiphysics simulations and experimental characterization in gas diffusion electrodes with a well-controlled, AAO-based microstructure. In doing so they resolve the impact of Li^+ and O_2 transport on Li_xO_2 morphology/distribution and the resulting electrochemical performance of Li- O_2 batteries. The authors propose that, by adjusting the Li^+ concentration in the electrolyte, the maximum allowable discharge capacity of the oxygen cathode can be due to either interfacial passivation due to an undesirable LiO_2 morphology, or poor reactant transport either due to pore blockage or low oxygen diffusivity/solubility. The experimental design seems to be well thought-out, and the implementation of AAO as a support to image deposition at various length scales is clever. I recommend publication of the manuscript after the following revisions are considered/addressed:

Comment #1: *In some cases, the clarity of the work would benefit from minor language editing. For example, the meaning of terms like “extreme point” and “tendency” (which I believe correspond to “global maxima” and “trend”?) are relatively difficult to parse in the contexts of which they are used. Unfortunately, I find that this does negatively impact my ability to understand the main conclusions of the work, which are well-communicated in the Fig.s but less so in the text.*

Response: We are sorry for the inconvenience caused by these terms. In this work, the physical meaning of “extreme point” is the Li^+ ion concentration leading to maximum capacity or maximum ionic conductivity. To avoid ambiguity, we have removed the term "extreme point" and revised the corresponding statement as follows: “Ionic conductivity is commonly considered a general explanation for capacity variations, with the reported peak conductivity of LiTFSI/TEGDME occurring between 1 M and 2 M. However, the Li^+ ion concentration for maximum capacity does not coincide with that for peak ionic conductivity” (**Page 5, lines 21-25**). The term “tendency” has been changed to “trend” in the revised manuscript. We have also made efforts to optimize the language editing and highlight them in the revised manuscript.

Comment #2: *Why are units of areal capacity and specific capacity used when comparing the disordered and AAO air electrodes? This seems relatively counter-intuitive and a rationale should be provided.*

Response: We thank the reviewer for this important comment. The choice of current density is a well-considered decision; however, this process was inadvertently omitted during manuscript editing, and we sincerely apologize for this inconvenience.

Due to the substantial structural differences between visualized electrodes (C-AAO electrodes) and disordered electrodes (CNT electrodes), it is deemed unfair to benchmark them solely based on areal or specific current. To address this disparity, we recalibrate the applied current for the various electrodes using the actual electrochemical active area (A_e) as a reference. Although the total surface area of the

electrode ($A_{total} = A_e + A_n$) can be determined through the BET method, it encompasses areas that do not actively partake in the electrochemical processes (A_n). Therefore, a double-layer capacitance (C_{dl}) method is used. CV curves of the different electrodes in the Ar atmosphere were measured, as shown in **Fig. R2a-b**. The voltage region is 2.90-2.96 V and the scan rates are 0.2-1.0 mV s⁻¹. C_{dl} is obtained by fitting a function of response current density and scan rate, as shown in **Fig. R2c-d**. Since the C_{dl} value is linearly proportional to the electrochemically active surface area of the electrode, the C_{dl} ratio of the C-AAO electrode and CNT electrode corresponds to their ratio of electrochemical active areas. As a result, the ratio = $C_{dl, CNT}/C_{dl, AAO} = 22.5 \text{ mF cm}^{-2}/5.07 \text{ mF cm}^{-2} = 4.44$. For the C-AAO electrode, the specific current is $i_{specific, AAO} = 300 \text{ mA g}^{-1}$ and the areal current is $i_{areal, AAO} = 0.03 \text{ mA}/(1.3 \text{ cm})^2/3.14 \times 4 = 0.0226 \text{ mA cm}^{-2}$. According to the C_{dl} ratio of the two electrodes, the areal current of the CNT electrode is $i_{areal, CNT} = 0.0226 \text{ mA cm}^{-2} \times 4.44 = 0.100 \text{ mA cm}^{-2}$. The parameters of the two electrodes' structure and applied current are summarized in **Table R2** for reader comprehension. The selection of operating protocol has been included in the revised manuscript (Page 5, lines 14-18) and revised Supplementary Information (Fig. S2 and Table S1).

Comment #3: *When comparing the impedance behavior of the 1 M and 0.1 M electrolytes before and after discharge, the net change in R_s and R_{ct} at the same capacity (i.e. the same mass of Li_2O_2 formed) should also be investigated to ensure that the differences are actually a result of the electrolyte concentration.*

Response: We thank the reviewer for this important suggestion. To provide a fair and objective comparison, the EIS results with a fixed capacity using both disordered and visualized electrodes in electrolytes with various Li^+ ion concentrations are added. Further, the corresponding SEM images are also supplemented to enhance understanding of impedance behaviors.

The fixed capacities of 1.5 mAh cm⁻² and 4000 mAh g⁻¹ are selected, respectively, for the disordered electrode and visualized electrode. The fixed capacity values are chosen before the rapid decline in voltage begins. The pristine and discharged EIS curves are shown in **Fig. R11**, and the net change in impedance is summarized in **Fig. R12** and **Table R1**. The net R_{ct} reaches a maximum in the 0.1 M electrolyte (2-4-fold) while remaining at lower and comparable levels in the 0.5-2 M electrolytes. R_s exhibits negligible growth in the 0.5-2 M electrolytes but increases substantially in the 0.1 M electrolyte (20-fold). The SEM images of the Li_2O_2 morphologies under different Li^+ ion concentrations are shown in **Figs. R9-R10**. The EIS results are in excellent agreement with the morphological evolution of Li_2O_2 , transitioning from a Li_2O_2 film at 0.05-0.1 M with severe electrode passivation to Li_2O_2 particles within the 0.5-2 M range.

The new EIS results, SEM images, and the discussion on the correlation between impedance behaviors and product morphologies have been included in the revised manuscript (lines 12-16 on page 6, lines 7-8 on page 7, **Fig. 1g-i**, and **Table S2**) and revised Supporting Information (**Fig. S6** and **Fig. S8**).

Fig. R11. The pristine and discharged EIS results at the fixed capacities of (a) disordered electrodes and (b) visualized electrodes.

Fig. R12. Comparison of the net R_s and net R_{ct} , where net R is defined as $R_{discharged} - R_{pristine}$.

Comment #4: *The limiting current measurements for Li^+ and O_2 cited by the authors are measured by glassy carbon RDEs and their relevance for gas diffusion electrodes are unclear. Furthermore, the values cited are very similar in magnitude and would be heavily dependent on the electrolyte chemistry (i.e. Li^+ concentration and O_2 solubility).*

Response: Our original intention was to emphasize that Li^+ ion is not the main limiting factor for mass transport. This conclusion can be drawn inferred from the difference in Li^+ ion and oxygen concentrations. For instance, in a 0.05 M electrolyte, the concentration of Li^+ ions is more than 10 times higher than that of oxygen. We agree with the reviewer's opinion that the relevance between the gas diffusion electrode and glassy carbon RDEs is unclear. Therefore, the reference to limiting current measurements has been removed in the revised manuscript.

Comment #5: *The nucleation-growth mechanism proposed by the authors seems to be more related to growth and has little to do with the nucleation regime, which should*

encompass the factors associated with initial Li_2O_2 seed formation and coverage before the transport limited regime. It is recommended that this regime either be discussed or the terminology be changed to something more akin to growth mechanism.

Response: We appreciate the reviewer's good suggestion which is helpful to improve the logic and readers' understanding of the proposed mechanisms. The nucleation-growth mechanism is discussed in two parts. **In the nucleation part**, we first propose “The nucleation of Li_2O_2 is distinct from the deposition of metallic Li or Zn in metal-based batteries”. Subsequently, we think that the initial nuclei density and the early voltage are determined by the adsorbed oxygen concentration on the electrode surface before discharge and electrode kinetics, as illustrated in **Fig. R13a**. Thus, more nuclei will be produced in electrolytes with low Li^+ ion concentration. **In the growth part**, a phase field model is used to characterize the Li_2O_2 growth in electrolytes with different Li^+ ion concentrations, based on which the effect of the nuclei density on the Li_2O_2 morphology and overpotential in the later stage of discharge is further quantitatively revealed.

The description of nucleation has been reorganized in the **revised manuscript (lines 23-29 on page 7, lines 1-5 on page 8)**: “In the LOBs, the overpotential undergoes a monotonic increasing process, since it arises from the kinetics from oxygen to superoxide ($\text{O}_2 + \text{e}^- \rightarrow \text{O}_2^-$), rather than a direct phase transition. The decrease in salt concentration enhances the oxygen solubility, while simultaneously freeing up sites on the electrode surface for oxygen molecules to adsorb before discharge. Therefore, a negative correlation between the initial voltage plateau and Li^+ ion concentration in Fig. 1d-f is observed (Fig. R13a) and the early voltage is controlled by the adsorbed oxygen (Fig. R13b). Additionally, more nuclei are generated at the beginning of discharge in the presence of higher oxygen concentration and faster electrode kinetics in electrolytes with low Li^+ ion concentration (Fig. R13a). The initial nuclei density will further determine the product morphology, thereby influencing the voltage characteristics in the later stage of discharge (Fig. R13b)”.

Fig. R13. (a) The mechanism of Li^+ ion concentration on initial voltage plateau and nucleation. (b) Scheme of voltage characteristics and control factors at early and later stages.

Comment #6: To confirm the author's proposed passivation film failure mechanism for the low- Li^+ concentration system, it would be useful for to provide data where the cell

is fully discharged, rested for sufficient time to regain reactant concentration at the interface, and then discharged again. If a passivating film is responsible for failure, one would expect there to be negligible capacity output after the rest step due to high interface impedance.

Response: We thank the reviewer for this suggestion. **Fig. R14** shows the voltage-capacity curves of a disordered electrode in the 0.05 M electrolyte undergoing two full discharges. Before the second discharge, the battery was rested for 72 hours. The current density is 0.1 mA cm^{-2} , consistent with that in Fig. 1. The capacity of the second discharge is only $0.014 \text{ mAh cm}^{-2}$, which can be negligible compared to the first discharge. The related discussion is included in the **revised Supporting Information (Fig. S9)**.

Fig. R14. The voltage-capacity curves of the disordered electrode in 0.05 M electrolyte undergoing discharge, rest, and discharge: (a) the first and second discharge; (b) Amplification diagram of the second discharge.

Comment #7: *The values for Li^+ and O_2 solubility and transport (e.g. diffusion coefficient) used in the electrolyte portion of the Multiphysics simulations should be explicitly stated, discussed, compared to the actual experimental systems, and rationalized on this basis in the main text.*

Response: We thank the reviewer for this important suggestion. The selection of values is well-founded and is determined based on experimental data from references and calibration of simulation results.

Due to differences in testing methods, these parameters lack uniform values even under the same LiTFSI concentration. Therefore, we summarized the measurement results from the current works to identify the basic trends in value variations. Subsequently, within the credible value range, we adjusted the values until the simulated voltage-capacity curves aligned with the experimental results. It is common practice in simulation methodology to infer the value of a parameter by matching experimental and simulated results.

Fig. R15a shows the oxygen solubility in TEGDME with 0-1 M LiTFSI is at a range of 3.4-5.6 mM under 1 atm oxygen pressure^{1,2,6,7}. A value of 4.43 mM is selected for the 0.5 M electrolyte. Regrettably, the solubility value for 2 M electrolyte has not

been reported. According to the matching degree of simulation and experiment, the solubility is determined to be 1.15 mM in this work. **Fig. R15b** shows the $-\lg(D_{O_2})$ from ref. ^{1,2,7-9}, which shows two linear relationship regions. The values of oxygen diffusivity used in the work are selected between the two regions, specifically $9.38 \times 10^{-11} \text{ m}^2 \text{ s}^{-1}$ for 0.5 M electrolyte and $2.35 \times 10^{-11} \text{ m}^2 \text{ s}^{-1}$ for 2 M electrolyte.

The diffusivity of Li^+ ion in the range of 0.5 M-2 M can be expressed by¹⁰

$$\frac{10^6 D_{\text{Li}^+}}{\text{cm}^2 \text{ s}^{-1}} = 1.821c_{\text{Li}^+}^2 - 9.051c_{\text{Li}^+} + 11.73 \quad (\text{Equation R1})$$

Thus, the values for the 0.5 M and 2 M electrolytes are $7.66 \times 10^{-10} \text{ m}^2 \text{ s}^{-1}$ and $0.912 \times 10^{-10} \text{ m}^2 \text{ s}^{-1}$. The above values are summarized in **Table R3**. To maintain text fluency, the discussion on parameter selection and rationality analysis is added in the **revised manuscript (Page 9, lines 8-10)** and **revised Supplementary Information (Fig. S11)**.

Fig. R15. The values of O_2 solubility and diffusivity at 1 oxygen pressure measured by experiments^{1,2,6-9} and used in the simulation of this work.

Table R3. The key parameters under different Li^+ ion concentrations used in this work

Li^+ ion concentration (M)	Oxygen solubility (mM)	Oxygen diffusivity ($\text{m}^2 \text{ s}^{-1}$)	Li^+ ion diffusivity ($\text{m}^2 \text{ s}^{-1}$)
0.5	4.43	9.38×10^{-11}	7.66×10^{-10}
2	1.15	2.35×10^{-11}	0.912×10^{-10}

Comment #8: Similarly, I cannot locate the simulation parameters for the 2 M Li^+ simulations. These must also be stated, and whether the aforementioned transport parameters were adjusted based on said concentration must be discussed.

Response: We are sorry for the inconvenience. The key parameters are adjusted according to the electrolyte concentration, as shown in **Fig. R15** and **Table R3**, and elucidated in response to **Comment #7**.

We thank the reviewer for the valuable comments, which are helpful for improving the quality of this manuscript.

Response to the Third Reviewer's Comments

This manuscript provides the correlation between nucleation and kinetic of the Li_2O_2 formation utilizing visualization methods such as C-AAO electrodes and computational modeling. Such an approach may interest researchers to understand the kinetical and structural challenges the LOBs have. However, the information provided here is based on a very limited situation and cannot represent the general phenomena happening in LOBs. Therefore, I suggest rejecting the publication and some questions for manuscript modification.

We thank the reviewer for taking the time to review our manuscript and the feedback provided is very useful to promote the novelty and generality of this manuscript. To address the reviewer's concerns, we have enhanced the discussion on the necessity of the C-AAO electrode and appended additional data to demonstrate the universality of the conclusions.

Firstly, we elucidate the necessity of C-AAO electrodes for the investigation of nucleation and transport kinetics:

(i) The C-AAO electrode allows for direct observation of the morphologies and distribution of Li_2O_2 within the whole electrode, including both surface and interior regions. Due to its brittleness, the C-AAO electrode can be easily bent to view the cross-section without destroying the distribution and morphology of Li_2O_2 inside the channels advantageously. In comparison, this is unachievable using common disordered electrodes, at least in the current research stage. As a result, most studies focus on the Li_2O_2 on the electrode surface rather than its interior. For the disordered electrodes with self-supported structures, two faces are visible (top and bottom). However, for the electrodes using carbon paper substrates with spray-coated active material, only one face is observable. To date, limited works have reported the states of discharge product inside the electrode, with findings remaining inconclusive. Bardenhagen et al. used Argon ion sputtering and XPS spectra to analyze the composition of the product at various depths¹¹. Dutta et al. assembled a coin cell with two stacked layered CNT electrodes¹², allowing access to information from three electrode faces. However, intricate details inside the porous electrode were overlooked, such as the morphology and distribution of products, and the state of the solid-liquid interface. Therefore, what happens in the interior of the electrode remains unknown. Research progress on the observation of products inside the gas electrodes is demonstrated in the Introduction section (**Page 4**). Currently, utilizing a functionalized electrode like a C-AAO electrode is an effective approach to unraveling phenomena inside the electrode (**lines 1-3 on page 9 in the revised manuscript**).

(ii) The highly consistent and controllable channel units of the C-AAO electrode overcome the randomness of disordered pores, thereby facilitating focused investigations into transport, nucleation, and growth phenomena. This approach clarifies and accurately maps the pathways for species transport, electrochemical reaction interfaces, species flux, and storage space for solid products. Importantly, the phase field model coupling multi-physical fields and the heterogeneous structure model can be better developed. In conclusion, the C-AAO electrode is a powerful tool for comprehensively understanding the coupling behaviors of transport and nucleation within the gas electrode of metal-gas batteries (**lines 3-8 on page 9 in the revised manuscript**).

Additionally, the universality of the conclusions drawn from the visualized electrode can be proved by the following efforts: (i) The selection of the operating protocols based on the actual electrochemically active area of the electrode. (ii) The remarkable consistency in the electrochemical and Li_2O_2 behaviors between the visualized and disordered electrodes. (iii) Rigorous proof of concept by tuning the operating protocol, electrode framework, and amplification experiment using general electrodes. The detailed processes are included in **Comment #1**.

Comment #1: Fig 1A and B show the discharge capacity with each electrode but they have different dimensions, which makes it hard to compare. Moreover, how can the phenomenon shown with a visualized electrode represent that shown with a disordered electrode?

Response: We thank the reviewer for this important question. Before commencing the experiment with C-AAO electrodes, (i) strict work conditions were carefully selected and (ii) the electrochemical and Li_2O_2 behaviors of the two electrodes were thoroughly evaluated. After the C-AAO electrode experiment, (iii) the conclusions were further validated using disordered electrodes. The considerations of the experiment design are illustrated in **Fig. R16**, and the details are as follows.

Fig. R16. The scheme of experiment design in this work.

(i) To address the adverse effects of structural differences, the applied current is determined based on the actual electrochemically active area (A_e). The double-layer capacitance (C_{dl}) method is used, which indicates the electrochemically active surface area of the CNT electrode (22.5 mF cm^{-2}) is 4.44 times that of the C-AAO electrode (5.07 mF cm^{-2}). The test results of C_{dl} are shown in **Fig. R2**. Consequently, the current densities based on A_e applied in the CNT electrode (0.1 mA cm^{-2}) and C-AAO electrode (300 mA g^{-1}) are the same, as listed in **Table R2**. Please refer to **Comment #2, Response to the First Reviewer's Comments** for detailed calculations (**Pages R3-R5**).

(ii) The electrochemical characteristics of the two electrodes at various Li^+ ion concentrations show a consistent trend, including discharge capacity, net changes in R_s and R_{ct} , and initial voltage plateau, as shown in **Fig. R1**. Additional experimental data under more conditions has been supplemented and compared, such as the Li_2O_2 morphologies and net impedance at the same capacity, as shown in **Figs. R10-R12**. These data indicate a high similarity in the behaviors of CNT electrodes and C-AAO electrodes. The consistency in electrochemical and Li_2O_2 behaviors between the two

electrodes is a prerequisite for the next research, and we have strengthened the discussion on this aspect in the **revised manuscript (Pages 5-6, lines 3-5 on page 7)**.

(iii) The conclusions drawn from the C-AAO electrode are rigorously validated using the CNT electrode (Fig. 5). The proof of concept section has been reorganized and divided into three steps in the **revised manuscript (Fig. 5 and pages 13-15)**. The details are as follows.

a) Tuning the operating protocol. Fig. R17a shows a shift in Li^+ ion concentration inducing the capacity peak from 0.5 M to 1 M when the discharge rates from 0.2 mA cm^{-2} to 0.05 mA cm^{-2} . As the rate decreases, the capacity of the 2 M electrolyte shifts from being below 0.1 M to above 0.1 M. This can be explained by the proposed theory, which is due to the weakening of the role of transport kinetics at lower rates and the less severe electrode passivation in electrolytes with high Li^+ ion concentration, as illustrated in Fig. R17b.

b) Tuning the electrode framework. A battery with a double layer of CNT electrode is assembled (Fig. R18a-b). The local transport ability of the electrodes is regulated by the position of the breathing channels with low tortuosity. Firstly, C-AAO experiments suggest that, in the 0.5 M electrolyte with good transport kinetics, the dense Li_2O_2 particles are predominantly located near the separator side (Fig. 3b, h, and j), leading to the clogging in that region and consequent discharge failure. To verify that, Fig. R18c-d shows that the capacity of the structure type I (better transport ability on the separator side) is 53% higher than that of type II (better transport ability on the oxygen side) in the 0.5 M electrolyte, which breaks the understanding of conventional electrode design. Secondly, C-AAO experiments also suggest that the cause of the discharge failure with a 2 M electrolyte is slow oxygen transport from external gas to electrode depth (Fig. 3c, f, g, and i). To verify that, Fig. R18c-d shows that the capacity of type II is 40% higher than that of type I in the 2 M electrolyte. The applied current densities for CNT electrodes are both 0.1 mA cm^{-2} . Therefore, the observed Li_2O_2 distribution inside the channels of the C-AAO electrode provides important groundbreaking insights for electrode design criterion. In return, the conclusions drawn from the C-AAO experiments are also validated successfully by this double-CNT experiment.

c) Amplification experiment. To further clarify the phenomena in b), more extreme conditions are employed, such as a double discharge rate (0.2 mA cm^{-2} and 0.5 M electrolyte) and poorer transport kinetics (0.1 mA cm^{-2} and 3 M electrolyte). As a result, under the condition of a double discharge rate, the effect of type I in the 0.5 M electrolyte is further boosted from 53% to 152% (Fig. R19a-b). With the poorer transport kinetics, the effect of type II is increased from 40% to 74% (Fig. R19a-b). These results further underscore the necessity for the universal design of electrode structures to align with the transport and nucleation kinetics.

Fig. R17. Tuning of the operating protocol. (a) The capacity trend with disordered electrodes under 0.05, 0.1, and 0.2 mA cm⁻². (b) Analysis scheme of the impact of discharge rates on capacity peak.

Fig. R18. Tuning of the electrode framework. (a) Scheme of the double-electrode structure. (b) SEM images of CNT and bre-CNT electrodes. (c) Galvanostatic discharge curves at 0.1 mA cm⁻² and (d) capacity trend of different structures using 0.5 M and 2 M electrolytes. Red represents structure type I and blue represents type II. Solid lines represent 0.5 M electrolytes and dotted lines represent 2 M electrolytes.

Fig. R19. Amplification experiment. (a) Galvanostatic discharge curves and (b) capacity trend of different structures with a double discharge rate and worse transport kinetics. Red represents structure type I and blue represents type II. Solid lines represent the condition of 0.5 M electrolytes and 0.2 mA cm⁻², and dotted lines represent the condition of 2 M electrolytes and 0.1 mA cm⁻².

Comment #2: *Fig 1H only provides the EIS results of the visualized electrode before discharge even rest of manuscript are based on the results utilizing C-AAO electrode. Results shown after discharge with C-AAO electrode should be attached.*

Response: We are very sorry for this inconvenience. This is an annotation error where Fig. 1h displays the result of the discharged C-AAO electrode.

To facilitate a fair and visual comparison of the impact of product morphology on impedance, the EIS results of CNT and C-AAO electrodes at the fixed capacities with various Li⁺ ion concentrations are supplemented. The fixed capacities are 1.5 mAh cm⁻² for the CNT electrode and 4000 mAh g⁻¹ for the C-AAO electrode. The criteria for selecting the capacity is to do so before the rapid voltage decay occurs. The pristine and discharged EIS results are shown in **Fig. R11**, and the net change in impedance is summarized in **Fig. R12** and **Table R1**. The net R_{ct} reaches a maximum in the 0.1 M electrolyte (2-4-fold) while remaining at lower and comparable levels in the 0.5-2 M electrolytes. R_s exhibits negligible growth in the 0.5-2 M electrolytes but increases substantially in the 0.1 M electrolyte (~20-fold).

In addition, SEM images of Li₂O₂ morphologies on the CNT and C-AAO electrodes are supplemented in **Figs. R9-R10**, which provides a clear explanation for impedance behaviors. For both CNT and C-AAO electrodes, Li₂O₂ exhibits a film-like structure in the 0.05-0.1 M electrolytes, and transforms into a particle-like structure in 0.5-2 M electrolytes.

The new EIS results, the corresponding SEM images, and the discussion on the correlation between impedance behaviors and product morphologies have been included in the **revised manuscript (lines 12-16 on page 6, lines 7-8 on page 7, Fig. 1g-i, and Table S2)** and **revised Supplementary Information (Fig. S6 and Fig. S8)**.

Comment #3: *Fig 3E and F show the oxygen flux with different concentrations of electrolyte. After discharge, it seems the electrode filled with particles more densely when 0.5 M was used but oxygen flux was much lower in 2 M. Explanation combined with SEM images should be provided.*

Response: In the manuscript, we stated on page 9 “The visible Li₂O₂ particles aggregate and fill the channels, as shown in **Fig. R20b**.” and “Interestingly, the channels are almost empty again when Li⁺ ion concentration further increases to 1 M and 2 M. The walls of the channel are covered with a layer of small particles, as shown in Fig. S14 and **Fig. R20c**”. Following this, the explanation for the SEM images is provided by the simulated results. In the next paragraph, we stated “The oxygen is the rate-limiting step in the electrochemical reaction for its concentration is much lower than

the Li^+ ion concentration. Higher viscosity reduces the ability to dissolve and transport oxygen, which is the primary factor influencing the discharge capacity and product morphology for the 0.5 M and 2 M electrolytes. For example, at 60% DOD, the minimum oxygen concentration in the 2 M electrolyte drops to 0.1 mM (Fig. R20f₁). In comparison, it is 35.8 times higher in the 0.5 M electrolyte (Figs. R20e₁ and R20f₁). Combined with the analysis of Sections 2.1 and 2.2, the maximization of discharge capacity in 0.5 M electrolyte can be attributed to the following factors: (i) a relatively high ion conductivity, reduces the ohmic resistance and charge transfer resistance; (ii) the nucleation mechanism for particle growth, prevents rapid electrode passivation and further mitigates polarization; (iii) the fast oxygen transfer characteristics, allows the electrode pore space to be efficiently utilized”. The lower oxygen flux in a 2 M electrolyte is determined by the electrolyte's ability to transport oxygen, such as its solubility and diffusivity. The solubility and diffusivity of oxygen in various Li^+ ion concentrations are provided in Table R3 and the analysis of the rationality of parameter selection is included on Page S15.

Furthermore, Li_2O_2 distribution is simulated in the 0.5 M and 2 M electrolytes to enhance the interpretation of the SEM images. The cases at different DODs are shown in Fig. R20h-j, while the dynamic processes of Li_2O_2 growth and oxygen consumption are captured in Movies S1-S2. By the synergy of visualization techniques and multi-physics field simulation, Li_2O_2 distribution and transport kinetics at different Li^+ ion concentrations are effectively explained.

Fig. R20. Quantitative analysis of species transport inside visualized electrodes under different Li^+ ion concentrations. (a) The cross-section of the C-AAO electrode with a channel diameter of 390 nm before discharge. The SEM images of Li_2O_2 distribution at the oxygen side, middle part, and separator side and the statistical analysis of particle size at the concentration of (b) 0.5 M and (c) 2 M. (d) The summary of average Li_2O_2 diameters with 0.5-2 M electrolytes. The simulated distribution of oxygen concentration at (e₁) 60%, (e₂) 80%, and (e₃) 100% DODs in the 0.5 M electrolyte and at (f₁) 60% and (f₂) 100% DODs in the 2 M electrolyte. (g) The average oxygen concentration and the corresponding voltage-capacity curves. The simulated distribution of Li_2O_2 volume fraction at (h₁) 60%, (h₂) 90%, and (h₃) 100% DODs in the 0.5 M electrolyte and at (i₁) 80% and (i₂) 100% DODs in the 2 M electrolyte. The positions of the Li_2O_2 maximum volume fractions are marked by white dots. (j) Migration of the Li_2O_2 maximum volume fraction site in the electrode depth direction as the discharge proceeds.

Comment #4: *In Fig 5D, the discharge capacity in each electrode configuration. The electrolyte concentrations are 0.5 M and 3 M, which makes the unity of the overall manuscript poor. Results with 2 M electrolyte should be contained.*

Response: We thank the reviewer for this comment which will be helpful to enhance the unity of the manuscript and reader comprehension. During the experimental design, the use of the 3 M electrolyte was considered carefully. In this section, we aimed to explore the regulation mechanisms for enhancing performance under different transport kinetics. Therefore, we opted for a 3 M electrolyte, which has poorer oxygen transport ability compared to the 2 M electrolyte, to amplify the experimental phenomena.

In the revised manuscript, we supplemented the results with 2 M electrolyte and 0.1 mA cm^{-2} . To strengthen the validation of the conclusion drawn from the C-AAO electrode, the double-CNT electrode experiment is reorganized and divided into two parts, including the (i) tuning the electrode framework and (ii) amplification experiment. In part (i), the battery with a double layer of CNT electrodes is discharged in 0.5 M and 2 M electrodes with 0.1 mA cm^{-2} (**Fig. R18**). The operating conditions are consistent with the Fig. 1, which provides a fair comparison. In part (ii), to amplify the phenomenon, more extreme conditions are employed, such as double discharge rate (0.2 mA cm^{-2} and 0.5 M electrolyte) and poorer transport kinetics (0.1 mA cm^{-2} and 3 M electrolyte), as shown in **Fig. R19**. This reorganization effectively enhances the unity of this work. The above is included in the **revised manuscript (Fig.5 and pages 13-14)**.

Comment #5: *Data in Fig 5D need to be compared with the cell assembled with single CNT and bre-CNT.*

Response: We thank the reviewer for this comment. The voltage-capacity curves of the single CNT and bre-CNT have been supplemented, as shown in **Fig. R21**. In this section, the purpose of assembling the battery with a double layer of disordered electrodes is to

demonstrate the effectiveness of optimization strategies tailored to varying degrees of nucleation and transport kinetics, rather than to compare the performance between dual-layer and single electrodes. Based on this flexible structure, it is easily feasible to modulate the local transport capability by changing the placement of the bre-CNT electrode. In the 0.5 M electrolyte with good transport kinetics and particle-nucleation mode, type I (bre-CNT placed close to the separator) shows a significant capacity enhancement compared to type II (Fig. R18). Thus, under the ideal transport and nucleation kinetics, addressing the pore blockage on the separator side to maintain oxygen transport is critical for breaking the capacity bottlenecks. While in the electrolytes with high Li^+ ion concentration, type II shows higher capacity (Fig. R18), for which the immediate priority is to enhance the rapid ingress of oxygen into the electrode. The performance of single CNT and bre-CNT has been included in the revised Supplementary Information (Fig. S21).

Fig. R21. The voltage-capacity curves of CNT and bre-CNT electrodes in the 0.5 M and 2 M electrolytes at 0.1 mA cm^{-2} .

We thank the reviewer for the valuable comments, which are helpful for improving the quality of this manuscript.

References

1. Haas, R. *et al.* Understanding the Transport of Atmospheric Gases in Liquid Electrolytes for Lithium–Air Batteries. *J. Electrochem. Soc.* **168**, 070504 (2021).
2. Gittleston, F. S., Jones, R. E., Ward, D. K. & Foster, M. E. Oxygen solubility and transport in Li-air battery electrolytes: Establishing criteria and strategies for electrolyte design. *Energy Environ. Sci.* **10**, 1167–1179 (2017).
3. Yang, L., Frith, J. T., Garcia-Araez, N. & Owen, J. R. A new method to prevent degradation of lithium-oxygen batteries: Reduction of superoxide by viologen. *Chem. Commun.* **51**, 1705–1708 (2015).
4. Gao, X., Chen, Y., Johnson, L. & Bruce, P. G. Promoting solution phase discharge in Li-O₂ batteries containing weakly solvating electrolyte solutions. *Nat. Mater.* **15**, 882–888 (2016).
5. Bergner, B. J., Schürmann, A., Peppler, K., Garsuch, A. & Janek, J. TEMPO: A mobile catalyst for rechargeable Li-O₂ batteries. *J. Am. Chem. Soc.* **136**, 15054–15064 (2014).
6. Wijaya, O. *et al.* A gamma fluorinated ether as an additive for enhanced oxygen activity in Li-O₂ batteries. *J. Mater. Chem. A* **3**, 19061–19067 (2015).
7. Schürmann, A. *et al.* Diffusivity and Solubility of Oxygen in Solvents for Metal/Oxygen Batteries: A Combined Theoretical and Experimental Study. *J. Electrochem. Soc.* **165**, A3095–A3099 (2018).
8. Laoire, C. O., Mukerjee, S., Abraham, K. M., Plichta, E. J. & Hendrickson, M. A. Influence of nonaqueous solvents on the electrochemistry of oxygen in the rechargeable lithium-air battery. *J. Phys. Chem. C* **114**, 9178–9186 (2010).
9. Mohazabrad, F., Wang, F. & Li, X. Experimental Studies of Salt Concentration in Electrolyte on the Performance of Li-O₂ Batteries at Various Current Densities. *J. Electrochem. Soc.* **163**, A2623–A2627 (2016).
10. Mehta, M. R., Knudsen, K. B., Bennett, W. R., McCloskey, B. D. & Lawson, J. W. Li-O₂ batteries for high specific power applications: A multiphysics simulation study for a single discharge. *J. Power Sources* **484**, 229261 (2021).
11. Bardenhagen, I., Fenske, M., Fenske, D., Wittstock, A. & Bäumer, M. Distribution of discharge products inside of the lithium/oxygen battery cathode. *J. Power Sources* **299**, 162–169 (2015).
12. Dutta, A., Ito, K. & Kubo, Y. Establishing the criteria and strategies to achieve high power during discharge of a Li-air battery. *J. Mater. Chem. A* **7**, 23199–23207 (2019).

Response to the First Reviewer's Comment

The manuscript has been well revised with all issues raised by the reviewers properly addressed. I would like to recommend it for publication in Nature Communication after addressing following minor suggestion.

Comment #1: *It is well known trade-off between capacity and cyclability in most of batteries particularly for lithium oxygen system. It is recommended for the authors to provide recharging properties of your cells that are redesigned to boost practical capacity and suggest some relevant discussions, which might help guide future research insights in this community.*

Response: We thank the reviewer for this important suggestion that will help enhance the depth of this work. A Li-O₂ battery operating under a full discharge-recharge protocol typically exhibits a limited cycle life, as the transport pathways for electrons, ions, and oxygen are significantly disrupted at the end of the discharge. To enhance cycle life, a fixed capacity is usually set during cycling. Operating the battery at a reduced fixed capacity helps preserve the species transport pathways, thereby extending cycle life. Generally, the fixed capacity is set at ~10% of the full discharge capacity (*Nature*, 2021, 592, 551–557; *Science*, 2023, 379, 499–505). Under the same electrode material and electrolyte conditions, achieving a high full discharge capacity means that more oxygen, ion, and electron pathways remain intact during fixed capacity cycling. Therefore, **increasing the full discharge capacity is crucial for not only improving the practical capacity (~10% of the full capacity) but also extending the cycling life.**

To demonstrate the feasibility of this approach, we supplemented the cycling performance of batteries assembled in type I and type II structures, operating at a current density of 0.2 mA cm⁻², as shown in **Fig. R1**. Fig. R1a-c shows cycling at a fixed capacity of 0.5 mAh cm⁻² (corresponding to 10% of the full discharge capacity for type I and 25% for type II), while Fig. R1d-f shows cycling at a fixed capacity of 1 mAh cm⁻² (20% for type I and 50% for type II). Due to the higher full discharge capacity of type I, it exhibits superior cycling stability compared to type II. **The above discussions have been included in the revised manuscript (lines 388-391 on Page 15) and revised Supplementary Information (Fig. S23).**

Additionally, the concentration of Li⁺ ions surrounding Li₂O₂ during charging could potentially influence the decomposition mechanism of solid discharge products. However, this aspect lies beyond the scope of this work and will be explored in our future work.

Fig. R1. Cycling performance of batteries assembled in type I (red) and type II (blue) structures under the current of 0.2 mA cm⁻² in the 0.5 M electrolytes with the fixed capacities of (a, b, c) 0.5 mAh cm⁻² and (b, e, f) 1 mAh cm⁻².

We thank the reviewer for the valuable comments, which are helpful for improving the quality of this manuscript.

Response to the Second Reviewer's Comment

I believe that the authors have sufficiently rectified the outstanding issues noted in the first round of review, and therefore recommend publication.

Response: We thank the reviewer for the positive comment on our work and your suggestions have helpfully promoted the quality of this work.

Response to the Third Reviewer's Comments

All questions were answered well enough. However, most of the findings in this work are not the next of currently established knowledge we have in LOBs. (e.g. oxygen gradient, salt concentration, or current density related to the morphology or formation of discharge products) The authors utilized C-AAO to observe the gas diffusion layer in cross-section, emphasizing it as a novel methodology, which has already been reported. Therefore, this work does not have enough novel points required for publishing Nature Communication. Belows are questions still remains.

Response: We are grateful to the reviewer for providing valuable feedback that will help us clarify our arguments. This work redefines the previously overlooked role of Li^+ ions in LOBs, emphasizing their critical importance. We demonstrate that **modulating the Li^+ ion concentration is essential for improving the nucleation and growth of Li_2O_2 , mass transport, and their interactions, leading to enhanced electrochemical performance.** Consequently, we establish a clear correlation between the microscopic behaviors and macroscopic performance under varying Li^+ ion concentrations, providing a new framework for optimizing LOB designs.

Table R1 summarizes some current knowledge of LOBs, including the performance mechanism with varying salt concentration, Li_2O_2 formation, Li_2O_2 distribution, and electrode design criteria. The main contributions of this work, in comparison to the established knowledge, are as follows:

(1) Li_2O_2 formation. The formation and morphology of Li_2O_2 are strongly correlated with Li^+ ion concentration. At low Li^+ ion concentration (0.05-0.1 M), higher adsorbed oxygen leads to a higher initial nuclei density, resulting in the formation of a Li_2O_2 film. As the Li^+ ion concentration increases (0.5-2 M), the nuclei density is reduced, causing Li_2O_2 to shift to a particle structure.

(2) Performance mechanism. The influence mechanism of Li^+ ion concentration on performance has been re-evaluated by introducing the Li_2O_2 behaviors. The early voltage is controlled by the adsorbed oxygen rather than direct phase transition, while the Li_2O_2 morphology and mass transport further influence the later voltage. The Li_2O_2 film formed at low Li^+ ion concentrations accelerates battery death. At high Li^+ ion concentrations, the key factor(s) leading to failure is (are) electrode blockage by Li_2O_2 particles or (and) poor oxygen transport.

(3) Li_2O_2 distribution. Li_2O_2 particle distributed against the oxygen gradient signifies a compatibility match for the nucleation and transport kinetics, enabling the output of the electrode's maximum capacity. The 0.5 M electrolyte causes the reversed Li_2O_2 distribution due to local surface deactivation near the oxygen inlet and sufficient oxygen supply deeper within the electrode. In contrast, at higher concentrations, Li_2O_2 distribution follows the oxygen gradient due to poor oxygen diffusion.

(4) Electrode Design. The electrode design should depend on the transport capability of electrolytes with different Li^+ ion concentrations and resulting Li_2O_2 distribution. For electrolytes with decent transport capability (0.5 M), addressing pore blockage on the separator side is the key to breaking the capacity bottleneck. Conversely, for the electrolytes with slow oxygen transport (1-3 M), the priority is to enhance oxygen ingress into the electrode to overcome sluggish transport kinetics

Then, although the C-AAO electrode is a useful tool for studying nucleation and transport kinetics, it is not the main focus of this work but rather a part of the methodology. The fabrication method of the C-AAO electrode as the study purpose was reported in our previous work (*Nano Letters*, 2022, 22, 7527–7534), with which the effect of current was briefly explored. **The novel point of this work lies in reassessing the importance of Li⁺ ion regulation mechanisms, not in the C-AAO electrode method.** We are sorry for any confusion caused by the term "novel" in the "a novel synergy of visualization techniques and cross-scale quantification" in the abstract. To clarify, we have revised the wording and emphasized the importance of Li⁺ ion regulation mechanisms in the revised manuscript (**lines 26-27 on Page 2**).

Table R1. Summary of some current knowledge of Li-O₂ batteries

	Journal	Research Status	Summary
Li ₂ O ₂ Formation	Nat. Chem. 6, 1091–1099 (2014).	Li ₂ O ₂ morphology is determined by the LiO ₂ equilibrium ($\text{LiO}_2 \rightleftharpoons \text{Li}^+_{(\text{sol})} + \text{O}_2^-_{(\text{sol})}$). Li ₂ O ₂ film is formed in the solvents with a low donor number (DN), and toroidal or particle-like Li ₂ O ₂ is formed with a high DN.	Previous studies have researched the effects of solvents, anions, and additives on Li ₂ O ₂ formation, while the role of Li ⁺ ions in this process remains unknown.
	Nat. Chem. 7, 50–56 (2015)	The additive, H ₂ O, with a high accept number (AN) promotes the growth of Li ₂ O ₂ toroids.	
	PNAS 112, 9293–9298 (2015)	The anions with higher DN than that of the solvent can promote the formation of Li ₂ O ₂ with larger toroid structures.	
Performance Mechanism	ACS Omega 4, 20708–20714 (2019)	The Li ⁺ ion conductivity and viscosity influence the Li-O ₂ battery performances significantly, while the oxygen solubility has little effect.	The possible reasons for the difference in performance with varying Li ⁺ ion concentration is (are) ion conductivity or (and) oxygen solubility.
	J. Electrochem. Soc. 163, A2623–A2627 (2016)	The balance between the ionic conductivity and mass transfer determines the highest capacity.	
	J. Phys. Chem. C 120, 5949–5957 (2016)	Lower electrolyte concentrations provided superior performance in terms of oxygen reduction, oxygen evolution, and overall efficiency in Li-O ₂ batteries.	
Li ₂ O ₂ Distribution	J. Mater. Chem. A 7, 23199–23207 (2019)	Li ₂ O ₂ yield on the oxygen side is higher than that on the separator side. With an increase in oxygen reduction, and more Li ₂ O ₂ is deposited near the air inlet.	The Li ₂ O ₂ distribution has not yet reached a definitive conclusion: (1) Most studies suggest that oxygen is the dominant factor influencing Li ₂ O ₂ distribution;
	Joule , 3, 542–556 (2019) Electrochim. Acta 425, 140703 (2022)	Most simulation works suggested that Li ₂ O ₂ aggregates on the oxygen inlet side.	

	J. Power Sources 299, 162–169 (2015).	Li ₂ O ₂ in the mesopores tends to be formed more at the ends of the air electrode, while in the macropores, it tends to be homogeneous.	(2) Some studies noted the potential role of Li ⁺ ions, while the detailed mechanisms and interactions of multiple factors remain unclear.
	Nano Lett. 22, 7527–7534 (2022)	At low current rates, Li ₂ O ₂ is uniformly distributed; however, at higher rates, the Li ₂ O ₂ size on the separator side is larger than on the oxygen side.	
Electrode Design	Electrochem. Commun. 46, 111–114 (2014)	A cathode structure with a gradient pore distribution (pore size reducing from 500 nm on oxygen inlet to 100 nm on separator face) was fabricated, which had a higher capacity than that of the uniform porous cathode.	Oxygen is considered a primary factor limiting the battery performance, and a common point for electrode design is to enhance the oxygen transport by setting higher porosity at the oxygen inlet or by employing hierarchical pore (channels) structures.
	J. Power Sources 451, 227821 (2020)	The numerical simulation suggested that an initial porosity gradient (0.73-0.77), with high porosity on the oxygen inlet, can enhance the discharge capacity.	
	Adv. Energy Mater. 13, 2302816 (2023)	Micron-scale vertical channels that penetrate through the nanoscale pores of the electrode are introduced to promote the diffusion depth of oxygen.	

Comment #1: *The difference in applied current density in CNT and C-AAO will affect the electrolyte concentration gradient near the electrode surface during discharge. The discussion and additional data should be included, particularly discharge capacity.*

Response: At the onset of our experimental design, we accounted for electrode differences and intentionally chose the double-layer capacitance (C_{dl}) method to ensure that the currents applied to the electrochemical surfaces of CNT and C-AAO electrodes are the same. In the revised manuscript, we claimed that “**0.1 mA cm⁻² applied in disordered electrodes is equal to 300 mA g⁻¹ of C-AAO electrodes according to the C_{dl} method**”, meaning that the consumption/generation rates of species on the active surface are consistent. In addition, **the absolute discharge capacities of the CNT and C-AAO electrodes are similar**, which can address the reviewer’s concerns.

Below are our considerations and the conversion methods of current and capacity:

(i) The reason for using the C_{dl} method. To design the experiment effectively, it is crucial to ensure that the micro-surfaces of CNT and C-AAO electrodes experience the same current, as the reviewer pointed out. The current is typically expressed in mA g⁻¹ or mA cm⁻². However, due to the differences in material density and specific surface area between the CNT and C-AAO electrodes, applying the same mA g⁻¹ or mA cm⁻² values would result in different currents across the electrode surfaces. Therefore, it is inappropriate to calibrate these two structurally different electrodes using identical mA g⁻¹ or mA cm⁻² values.

The current should be recalibrated based on the material's surface area. The Brunauer-Emmett-Teller (BET) method is commonly used to measure the total surface area (A_{total}) through physical adsorption, but it includes areas that do not participate in electrochemical processes (A_n), which introduces inaccuracies. A more precise method is to measure the electrochemical surface area (A_e). To compare the catalytic properties of different nano-materials fairly for a given reaction, the ACS Nano editorial board published an initiative titled "Best Practices for Reporting Electrocatalytic Performance of Nanomaterials" (ACS Nano, 2018, 12, 9635–9638). They believed that A_e measurements enable more accurate and equitable comparisons of nano-material performance and have endorsed the use of the C_{dl} method.

(ii) The conversion method for the same current based on the C_{dl} method. Cyclic Voltammetry (CV) curves of the different electrodes in the Ar atmosphere were measured, as shown in **Fig. R2a-b**. The voltage region is 2.90-2.96 V and the scan rates are 0.2-1.0 mV s⁻¹. C_{dl} is obtained by fitting a function of response current and scan rate, as shown in **Fig. R2c-d**. Since the C_{dl} value is linearly proportional to the electrochemically active surface area of the electrode, the C_{dl} ratio of the C-AAO electrode and CNT electrode corresponds to their ratio of electrochemical surface areas. As a result, $C_{dl, CNT}/C_{dl, AAO} = 22.5 \text{ mF cm}^{-2}/5.07 \text{ mF cm}^{-2} = 4.44$. For the C-AAO electrode, the specific current is $i_{specific, AAO} = 300 \text{ mA g}^{-1}$ and the areal current is $i_{areal, AAO} = 300 \text{ mA g}^{-1} \times 0.1 \text{ mg} / (1.3 \text{ cm})^2 / 3.14 \times 4 = 0.0226 \text{ mA cm}^{-2}$. According to the C_{dl} ratio of the two electrodes, the areal current of the CNT electrode is $i_{areal, CNT} = 0.0226 \text{ mA cm}^{-2} \times 4.44 = 0.100 \text{ mA cm}^{-2}$. The parameters of the two electrodes' structure and applied current are summarized in **Table R2** for comprehension. The selection of current has been shown in the revised manuscript (**lines 107-110 on Page 5**) and revised Supplementary Information (**Fig. S2 and Table S1**).

(iii) The absolute capacities of the CNT and C-AAO electrodes are similar due to the considerable choice of currents. For a clear comparison, the absolute capacities of the two electrodes in the 0.05-2 M electrolytes are shown in **Table R3**. The capacities of the two electrodes exhibit the same trends, with values for the same Li⁺ ions concentration being proximate. Taking a 0.5 M electrolyte as an example, the conversion method is as follows: $32700 \text{ mA g}^{-1} \times 0.1 \text{ mg} = 3.27 \text{ mAh}$, $4.18 \text{ mAh cm}^{-2} \times 0.5024 \text{ cm}^{-2} = 2.08 \text{ mAh}$. The consistency in current and capacity has provided a good foundation for comparing Li₂O₂ morphologies and understanding mechanisms. The comparison of absolute capacities has been included in the revised manuscript (**lines 110-111 on Page 5**) and revised Supplementary Information (**Table S2**).

Fig. R2. The double-layer capacitance (C_{dl}) of the air electrode measured by a CV method. CV curves of (a) C-AAO electrode and (b) CNT electrode in the region of 2.90-2.96 V vs. Li at 0.2-1.0 mV s^{-1} scan rates. C_{dl} of (c) C-AAO electrode and (d) CNT electrode.

Table R2. The parameters of C-AAO electrode and CNT electrode

Parameters	C-AAO electrode	CNT electrode
Diameter	13 mm	8 mm
Weight	0.1 mg (carbon)	2.5 mg (carbon)
C_{dl}	5.07 mF cm^{-2}	22.5 mF cm^{-2}
Applied current	0.0226 mA cm^{-2} (300 mA g^{-1})	0.1 mA cm^{-2} (20 mA g^{-1})

Table R3. The absolute capacities of CNT and C-AAO electrodes using electrolytes with different Li^+ ions concentration

Li^+ ions concentration (mM)	Absolute capacity (mAh)	
	CNT electrode	C-AAO electrode
0.05 M	0.63	0.60
0.1 M	1.20	1.81
0.5 M	2.08	3.27
1 M	1.79	2.61
2 M	1.15	1.14

Comment #2: *Morphological differences when CNT and C-AAO are discharged at the same current density, which is extending question of 1).*

Response: Li_2O_2 morphologies on the CNT and C-AAO electrodes discharged at the same current have been shown in **Figs. R3-R4** (Fig. S6 and Fig. S8 in the Supplementary Information). The criteria for selecting the capacity is to do so before the rapid voltage decay occurs. In the electrolytes with low Li^+ ion concentrations (0.05 M and 0.1 M), Li_2O_2 appears as a film enveloping the surface of CNTs; while it presents as particles with higher Li^+ ion concentrations (0.5-2 M). This indicates that with the increase in Li^+ ion concentration, Li_2O_2 transitions from a film-like structure to a particle-like one.

Fig. R3. The SEM images of Li_2O_2 morphology on the disordered electrode at the fixed capacity of 1.5 mAh cm^{-2} using (a) 0.1 M, (b) 0.5 M, (c) 1 M, and (d) 2 M electrolytes.

Fig. R4. The SEM images of Li₂O₂ morphology on the visualized electrode at the fixed capacity of 4000 mAh g⁻¹ using (a) 0.05 M, (b) 0.1 M, (c) 0.5 M, and (d) 2 M electrolytes.

We thank the reviewer for the valuable comments, which are helpful for improving the quality of this manuscript.

Response to the Third Reviewer's Comments

The authors have responded to all questions properly and this work has done systematically reassessed compared to previous works. Therefore, I recommend the publication to the nature communication. However, since what was mainly shown in this manuscript was a type of model electrode called AAO, it is not positive in terms of whether it is really useful for lithium-air batteries, which are actually being studied. I hope the author will conduct follow-up research on this in the future.

Response: We appreciate the reviewer's positive feedback. Your concerns will be addressed in our future work, and we invite you to stay tuned for our upcoming developments.